# Mechanically robust neuroprotective stent by sequential Mg ions release for ischemic stroke therapy

Hongkang Zhang [1,8], Yang Zhang [2,3,4,8], Lili Sheng[5,8], Xiaofeng Cao[1], Chuanjie Wu[2,4], Baoying Song[4], Yunong Shen[1], Zikai Xu[6], Ge Song[5], Hao Sun[5], Qing Liu[5], Xunming Ji [2,3,4] ✉, Miaowen Jiang[3] ✉, Ming Li [2,3] ✉ & Yufeng Zheng [1,7] ✉

Most acute ischemic stroke patients with large vessel occlusion require stent implantation post-thrombectomy for complete recanalization, yet they exhibit a high rate of poor prognosis due to ischemia-reperfusion injury. Thus, combining reperfusion therapy with neuroprotective treatment offers significant advantages. This study introduces a novel $Mg^{2+}$ eluting stent by incorporating neuroprotective $MgSO_4$ particles into a PLCL (poly (l-lactide-co-ε-caprolactone)) substrate using 3D printing technology. A novel $MgSO_4$-particle/ $Mg^{2+}$-ions combined-mechanical reinforcement mechanism was introduced. Subsequently, the neuroprotective efficacy of the stents was validated through oxygen-glucose deprivation/reoxygenation-injured neuron cells in vitro and via the transient middle cerebral artery occlusion rat model to emulate human brain ischemia/reperfusion injury in vivo. The staged-release of $Mg^{2+}$ is supposed to provide sequential neuroprotection that aligns with the treatment window for acute ischemic stroke. This study marks the first development of biodegradable neuroprotective brain stents and presents an effective strategy to alleviate cerebral ischemia-reperfusion injury.

Large vessel occlusion caused by intracranial atherosclerosis accounts for 15-35% of acute ischemic stroke (AIS) cases, leading to high morbidity and mortality worldwide[1,2]. Stent placement after thrombectomy is often necessary for complete revascularization in these patients, despite the associated risk of ischemia-reperfusion injury during the acute phase[3–5]. Therefore, the integration of neuroprotection and stent placement could be a promising strategy. However, most of the biomaterial scientists and manufacturers have only focused on the vascular tissue and blood compatibility of stent materials, as well as the mechanical optimization of stent structures.

The impact of dissolved ions or degradation product on brain tissue has not been studied yet, but our previous research has confirmed the correlation between them[6]. Biodegradable stents, representing a new generation, offer temporary scaffolding that dissolves, thereby restoring normal functions[7,8]. Consequently, an innovative concept of biodegradable neuroprotective stents has been introduced.

Cerebral stents should be biomechanically compatible with the brain vasculature. Drawing on research into biodegradable cardiovascular stents, biodegradable brain stents are garnering increased attention[9], which can be categorized into biodegradable metallic and

[1]School of Materials Science and Engineering, Peking University, Beijing, China. [2]China-America Institute of Neuroscience and and Beijing Institute of Geriatrics, Xuanwu Hospital, Capital Medical University, Beijing, China. [3]Beijing Institute of Brain Disorders, Capital Medical University, Beijing, China. [4]Department of Neurology and Neurosurgery, Xuanwu Hospital, Capital Medical University, Beijing, China. [5]Beijing Advanced Medical Technologies, Ltd. Inc, Beijing, China. [6]School of Life science University of Glasgow, Scotland, UK. [7]Faculty of Advanced Science and Technology, Kumamoto University, 2-39-1 Kurokami, Chuo-Ku, Kumamoto, Japan. [8]These authors contributed equally: Hongkang Zhang, Yang Zhang, Lili Sheng. ✉e-mail: jixm@ccmu.edu.cn; jiangmiaowen415@163.com; liming@xwhosp.org; yfzheng@pku.edu.cn

polymeric stents. Compared with biodegradable metals, polymers can be more readily customized into various structures and shapes as needed[10–15]. This affords polymer stents greater adaptability to the intracranial blood vessels. However, due to the inferior mechanical properties of polymers, enhancing the mechanical performance of polymer stents has become a major challenge for their application as cerebrovascular stents[11].

Regarding the neuroprotective strategy to alleviate ischemic reperfusion injury, neuroprotective agents within stents are designed to provide initially rapid release within hours, followed by days of sustained release. This is crucial because the timing of treatment is pivotal for addressing acute ischemic stroke, encompassing acute, subacute, and chronic stages. The acute stage features extensive cell death, destruction of the blood-brain barrier, reactive oxygen species-specific inflammatory response, and secondary damage from the spread of inflammation[16]. The subacute phase is another critical phase with upregulation of pro-repair mechanisms, which are vital for neurological prognosis[17,18]. Therefore, ideal neuroprotective stents may offer sequential neuroprotection to remote ischemic brain tissues, providing timely prevention of neuronal death during the acute stage and long-term enhancement of functional recovery from neural injury.

The neuroprotective effect of $Mg^{2+}$ has been widely confirmed in numerous studies, particularly in treating cerebral ischemic diseases[19]. Three main mechanisms are proposed to explain the neuroprotective function of magnesium: inhibiting intracellular calcium overload, stabilizing brain energy metabolism, and enhancing cerebral blood reperfusion[20–22]. Clinical studies indicate that patients with high serum magnesium levels exhibit a better prognosis for brain injury[23], while patients with acute cerebral ischemia and high brain magnesium content sustain less neuronal damage compared to those with low levels[24]. Treatment can be administered via intraperitoneal, intravenous, arterial, and intracranial routes[25]. When administered to the carotid arteries, low doses rapidly elevate drug concentration in the target area. Studies have shown that injecting magnesium directly into target organs via proximal arteries minimizes side effects and maximizes neuroprotection[26]. Mg-based functional fillers have been incorporated into PLCL or PLGA (poly (lactic-co-glycolic acid)) polymers as Mg, MgO, and $Mg_3(PO_4)_2$ for osteogenesis[27–34].

In this work, $MgSO_4$ dissolves readily in blood and body fluids, eliciting efficient biological responses. As depicted in Fig. 1, in this study, $MgSO_4$ particles, serving as inorganic reinforcement fillers, were added to the stent PLCL matrix using 3D printing technology to potentially enhance the stent's strength and provide beneficial effects for brain disease therapy. The chemical, mechanical, and biological properties of the stent were first evaluated in vitro. Subsequently, the stent material was implanted into the common carotid artery (CCA) of a rat model with middle cerebral artery occlusion to observe the in vivo neuroprotection offered by the stents.

## Results

### Characterizations of PLCLxMS stent materials

Scanning electron microscope (SEM) images in Fig. 2A and Fig. S1 illustrate that, when the $MgSO_4$ content was below 10 wt.%, the surface topography of the composite stents displayed a smooth matrix with uniformly embedded particles of a few microns in diameter, in contrast to the pure PLCL stent. Subsequent energy dispersive

### 3D Printed Biodegradable Neuroprotective stent

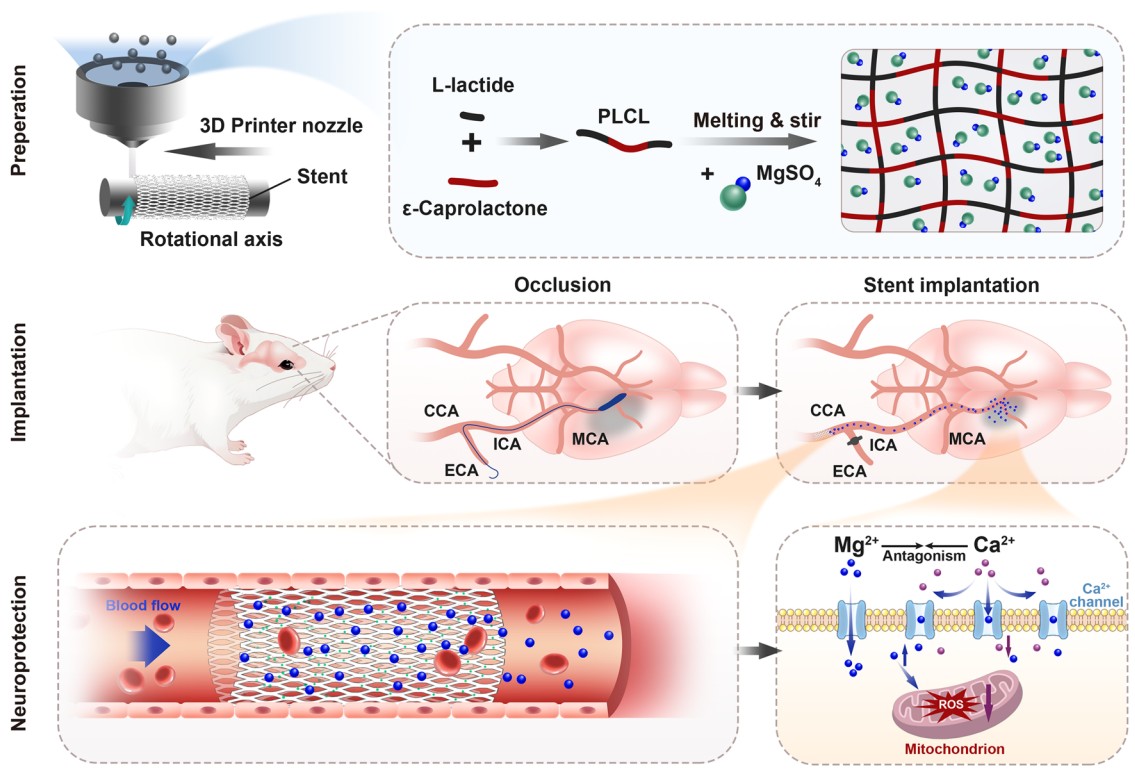

**Fig. 1 | Schematic illustration of the study design.** Preparation: PLCL is synthesized via copolymerization of L-lactide and ε-caprolactone. Magnesium sulfate ($MgSO_4$) powder is mixed into the molten PLCL to form PLCL/$MgSO_4$ composites with varying $MgSO_4$ concentrations. The composite is extruded as a hot-melt filament through a 3D printer nozzle, adhering to either a rotating rod or previously deposited filaments, ultimately fabricating stents. Implantation: A middle cerebral artery occlusion (MCAO) rat model is established. The fabricated stents are implanted into the common carotid artery (CCA). Neuroprotection: $Mg^{2+}$ ions (blue spheres) released from the stent migrate with blood flow into the infarcted region (gray area). There, they antagonize $Ca^{2+}$ overload and scavenge reactive oxygen species (ROS) in damaged neurons, exerting a neuroprotective effect.

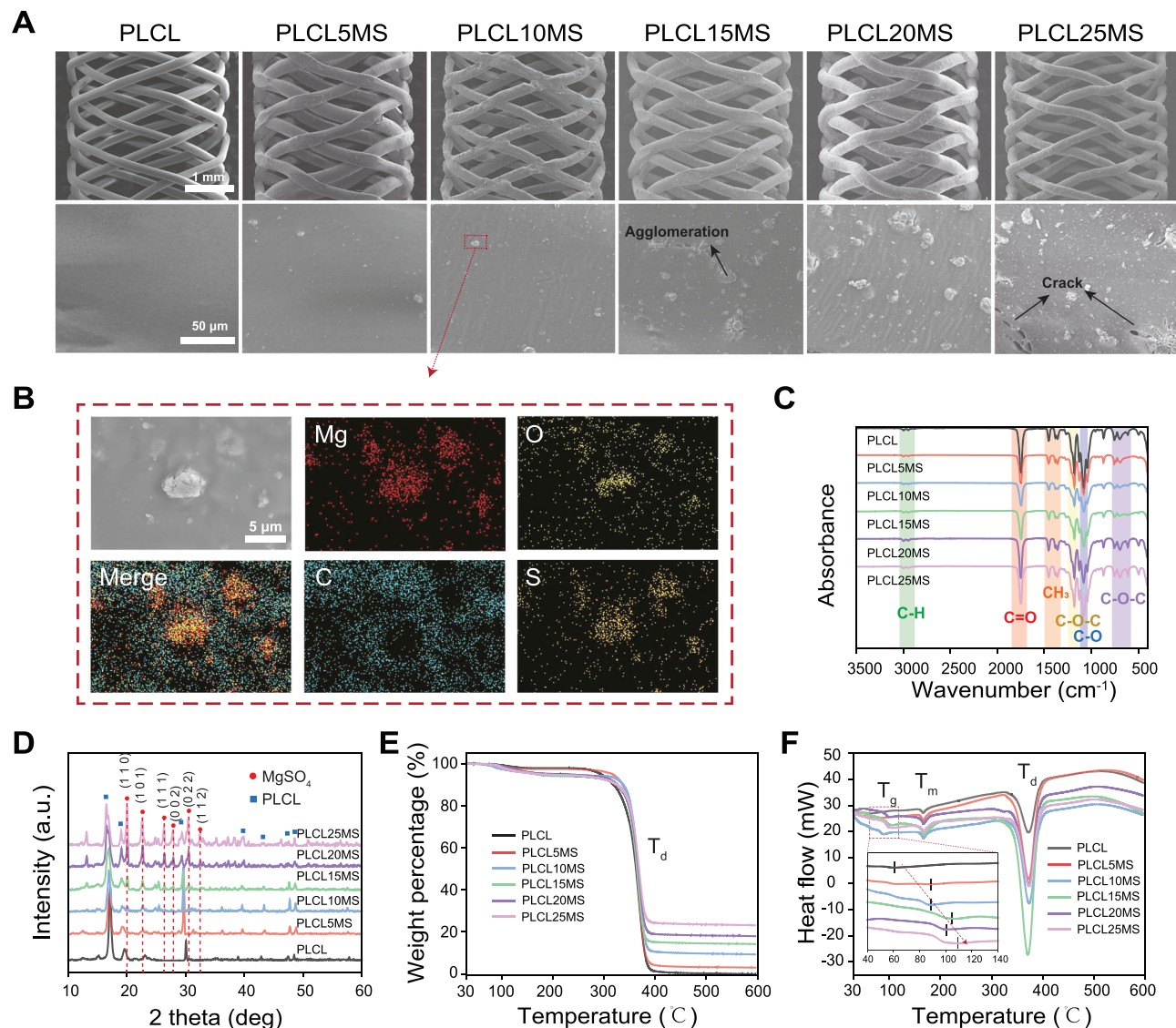

**Fig. 2 | Characterization of the PLCLxMS stents. A** Surface morphology of PLCLxMS (x = 0, 5, 10, 15, 20, 25) stents (SEM images). Each experiment was repeated 3 times independently with similar results. **B** Elemental mapping on the PLCL10MS stent surface. Each experiment was repeated 3 times independently with similar results. **C** Fourier transform infrared spectroscopy (FTIR) analysis. **D** X-ray diffractometer (XRD) spectra. **E** Differential scanning calorimeter (DSC) curves. **F** Thermogravimetric analysis (TGA) thermograph. Source data are provided as a Source Data file.

spectrometer (EDS) mapping in Fig. 2B confirmed that the particles were primarily composed of MgSO₄. Particle agglomerations became particularly noticeable when the MgSO₄ content exceeded 10 wt. %, attributed to the high surface energy and reduced distance between the particles[35]. Fig. 2C shows the Fourier Transform Infrared Spectroscopy (FTIR) spectrum of the stents, where typical peaks of -CH₃, C-O-C, O-H (free), and C=O from PLCL were identified. The addition of MgSO₄ particle did not alter the position of these characteristic peaks, suggesting that the interaction between MgSO₄ particles and the PLCL matrix involves physical binding. Following X-ray diffractometer (XRD) analysis in Fig. 2D confirmed the presence of MgSO₄ in the composites. Contact angle testing (Fig. S2) reveals that the hydrophilicity of the PLCL polymer increases gradually with the addition of MgSO₄, as evidenced by the decrease in contact angle. This increase in hydrophilicity can be attributed to the water-soluble property of MgSO₄ and the increased surface roughness observed in composites with higher MgSO₄ content (Fig. 2A)[36]. The heat flow curves of the composites for the first heating after standard and partial enlargements are plotted in Figs. 2E, F, respectively. Three thermal transitions are observed in all

samples: the glass transition of PLCL ($T_g$), the melting peak ($T_m$) and the degradation temperature ($T_d$). The $T_g$ shifted toward higher temperatures from 70°C to 100 °C with the increase of MgSO₄ content. The single $T_g$ indicated that both PLCL and the composites were initially completely amorphous[37]. The shift in $T_g$ could be attributed to the presence of MgSO₄ particles, which absorbed heat during the heating processes and hinder the polymer chains, making them harder to relax[38-40]. Furthermore, all samples exhibited $T_m$ at 150 °C and $T_d$ at approximately 360 °C, both of which remained constant across all standard extruded composites, indicating that the addition of magnesium sulfate did not affect the crystallinity of PLCL[37]. The weight loss as a function of temperature, obtained from thermogravimetric analysis (TGA), is depicted in Fig. S3 for the precision extruded composites with varying weight fractions of magnesium sulfate particles.

**In vitro degradation analysis**
To elucidate the Mg²⁺ release mechanism from PLCLxMS stents, we designed an experimental approach combining in vitro degradation studies with substrate degradation simulation. The Mg²⁺ release

diffusion model, developed based on the in vitro immersion release profile, is presented in Fig. 3A. For verification purposes, we established a computational model that incorporates the proposed release mechanism and aligns with the diffusion model parameters. $Mg^{2+}$ release and degradation of the PLCL in PBS is divided into three stages (Burst, Fast, and Stable), as depicted in Fig. 3A. This model integrates findings from previous research on the degradation characteristics of PLCL[41,42] and this study. In the first stage, within hours, there is an explosive release of $Mg^{2+}$ from the stent into PBS due to the dissolution of $MgSO_4$ embedded in the stent's surface. In the second stage, about one week, the concentration of $Mg^{2+}$ continues to rise but at a reduced rate. This stage features in the penetration of water into the PLCL matrix and the outward diffusion of $Mg^{2+}$ from within the stent. In the third stage, all $MgSO_4$ within the stents has dissolved and been released, while the PLCL chains begin hydrolysis, breaking down into oligomers and monomers. The key characteristics of each stage in the $Mg^{2+}$ release profiles are summarized in Table S1.

As illustrated in Fig. 3B, C, the release of $Mg^{2+}$ in PBS exhibits three distinct stages: Burst, Fast, and Stable. The disparity in magnesium release rates among these stages was striking, with the most rapid release occurring in the "Burst" stage, reaching tens of $mg \cdot L^{-1} \cdot day^{-1}$, followed by a comparatively slower release rate in the "Fast" stage, around a few $mg \cdot L^{-1} \cdot day^{-1}$. In contrast, the release rate of $Mg^{2+}$ diminished further in the "Stable stage", dropping to less than $1 \, mg \cdot L^{-1} \cdot day^{-1}$. Despite lasting only 1 day, in the "Burst" stage, the concentration of released $Mg^{2+}$ reached more than half of the total released over the 56-day period, indicating its explosive release characteristic. Additionally, the concentration of released $Mg^{2+}$ increased with higher $MgSO_4$ content in the stents. As shown in Fig. 3D and Fig. S4, the $MgSO_4$ particles embedded in the stent surface were completely dissolved after 7 days of immersion, leaving behind cavities at their original locations. However, partly $MgSO_4$ particles remained deep inside the stent struts obtained from the cross-sectional morphology, this corresponding to the "Fast" stage. Because the rate of penetrating of water into the PLCL matrix is a fast and then slow process[41], after the surface layer of magnesium sulfate is dissolved, the shallow magnesium sulfate particles will come into contact with water molecules as the water is penetrating, thus dissolving and releasing magnesium ions. These magnesium ions are released into solution by outward diffusion. The rate of magnesium ion release also decreases during this phase as restricted by the rate of penetration of water. The reported complete degradation time of PLCL stent in vivo is about 12 months, with a rapid degradation rate occurring by the sixth month[43], corresponding to the "Stable" stage. The dissolution and release of $Mg^{2+}$ from the $MgSO_4$ particles within the stent may occur through a diffusion process in this study, leading to no additional burst release of $Mg^{2+}$, but instead a slow-release rate. The weight loss of PLCLxMS stents, as shown in Fig. S5, increased proportionally with higher $MgSO_4$ content. This trend primarily arises because, while PLCL stents undergo minimal short-term weight loss during biodegradation, the dissolution of $MgSO_4$ significantly contributes to the overall weight loss. During the initial 7 days of immersion, there was a notable decrease in solution pH (Fig. S6). This was primarily attributed to the dissolution of water-soluble $MgSO_4$ embedded in the surface (30 g/100 ml at 20 °C) and the hydrolysis of $Mg^{2+}$ ($Mg^{2+} + 2H_2O = Mg(OH)_2 + 2H^+$) (1). However, after 7 days of immersion, the concentration of $Mg^{2+}$ in the PBS solution continued to increase, albeit at a slower rate, while the pH decreased.

Subsequent simulations of dissolution and degradation align well with the experimental results of the cumulative $Mg^{2+}$ release curve, further demonstrating that the release of $Mg^{2+}$ from the composite material adheres to the diffusion model extensively discussed in prior research[41]. Figure 3E illustrates the simulation of Mg concentration changes in the composite and solution over time during the degradation process. Initially, the magnesium concentration (red) on the surface of the composite material decreases rapidly, corresponding to

the release of magnesium ions during the first 7 days after immersion, as noted in Fig. 3C. Subsequently, $Mg^{2+}$ within the bulk gradually releases into the solution and diffuses. Eventually, the magnesium concentration becomes uniform throughout the entire model, with all $MgSO_4$ dissolved into $Mg^{2+}$ and uniformly distributed. Fig. 3F presents the cumulative release curve of $Mg^{2+}$ by simulation, following the same trend as Fig. 3D. The high congruence between experimental and simulation results indicates that the release of magnesium ions is dominated by the diffusion process.

## Mechanical characterization

**Mechanical effects of $MgSO_4$ particles in PLCLxMS.** As depicted in Fig. 4, we measured the radial strength of the PLCLxMS stents and the Young's modulus of the composites through sheet tensile tests. PLCL5MS stents showed the highest ultimate tensile strength (UTS), reached $51.3 \pm 1.85$ MPa, more than twice the UTS of pure PLCL sheets (Fig. 4B). And the mechanical properties of PLCLxMS sheets exhibited a significant increase in tensile modulus with the increase in $MgSO_4$ content (Fig. 4C), which is consistent with the Halpin-Tsai theoretical model widely utilized for estimating the modulus of nanofiller-reinforced composites[44–46]. However, the PLCL25MS composites demonstrated high brittleness, as evidenced by an elongation at break of only about 1% (Fig. 4B). This brittleness was attributed to cracks that formed near large-sized $MgSO_4$ agglomerations (Fig. 4D)[30]. After being fabricated into stents, the highest radial strength of the PLCLxMS stents (Fig. 4E–G) reached $1.36 \pm 0.159$ N/mm, also more than twice that of pure PLCL stents ($0.544 \pm 0.0825$ N/mm), due to the $MgSO_4$ particle strengthening effect[47,48]. Adequate mechanical support typically prevents acute recoil and strut rupture after implantation[49], enhancing the efficacy of PLCLxMS stents. However, the radial strength of the composite stents decreased with an increase in $MgSO_4$ content, with the radial strength of PLCL20MS and PLCL25MS stents even lower. This decrease could be attributed to poor interfacial bonding between large agglomerations and PLCL matrix, resulting in ineffective stress transfer across the matrix-filler interface during mechanical loading[50]. Thus, there is a trade-off between high radial strength and efficient magnesium ion release, both influenced by the $MgSO_4$ content.

Simulation results based on the stent plasticity model are shown in Fig. 4H. The greater the percentage of compression of the stents, the greater the $F_v$, and there is a concentration of stress at the intersection. Among the composites, the stress of PLCL5MS was the highest at the same compression ratio. Subsequently, the stress decreased with the increase in $MgSO_4$ content. The stress-strain curves obtained from the simulation (Fig. 4I) are well-fitted to the experimental results, validating our simulation model, which incorporates reaction engineering and ion diffusion.

## $Mg^{2+}$ strengthening mechanisms during PLCLxMS degradation

The change in mechanical properties of PLCLxMS stents during degradation was analyzed through radial force tests, as shown in Fig. 5. The radial force of all composite stents decreased gradually over immersion time (Fig. 5A) This decrease was mainly concentrated in the "Stable" stage, while there was only a very small decline in the "Burst" and "Fast" stages, which indicated that the dissolution of embedded $MgSO_4$ particles in surface did not cause a significant decrease in the mechanical strength of the stents. Interestingly, only pure PLCL stents became brittle after 14 days of immersion, losing their plasticity, and showed significant fracture in compression tests with the stent diameter compressed to about 2.14 mm (Fig. S7). This phenomenon is attributable to the properties of PLCL copolymers, which undergo a transition to glassy plastics during hydrolytic degradation[51,52]. In contrast, no fractures were observed in PLCLxMS composite stents, indicating that the addition of $MgSO_4$ is helpful for providing sufficient mechanical support of stents during degradation.

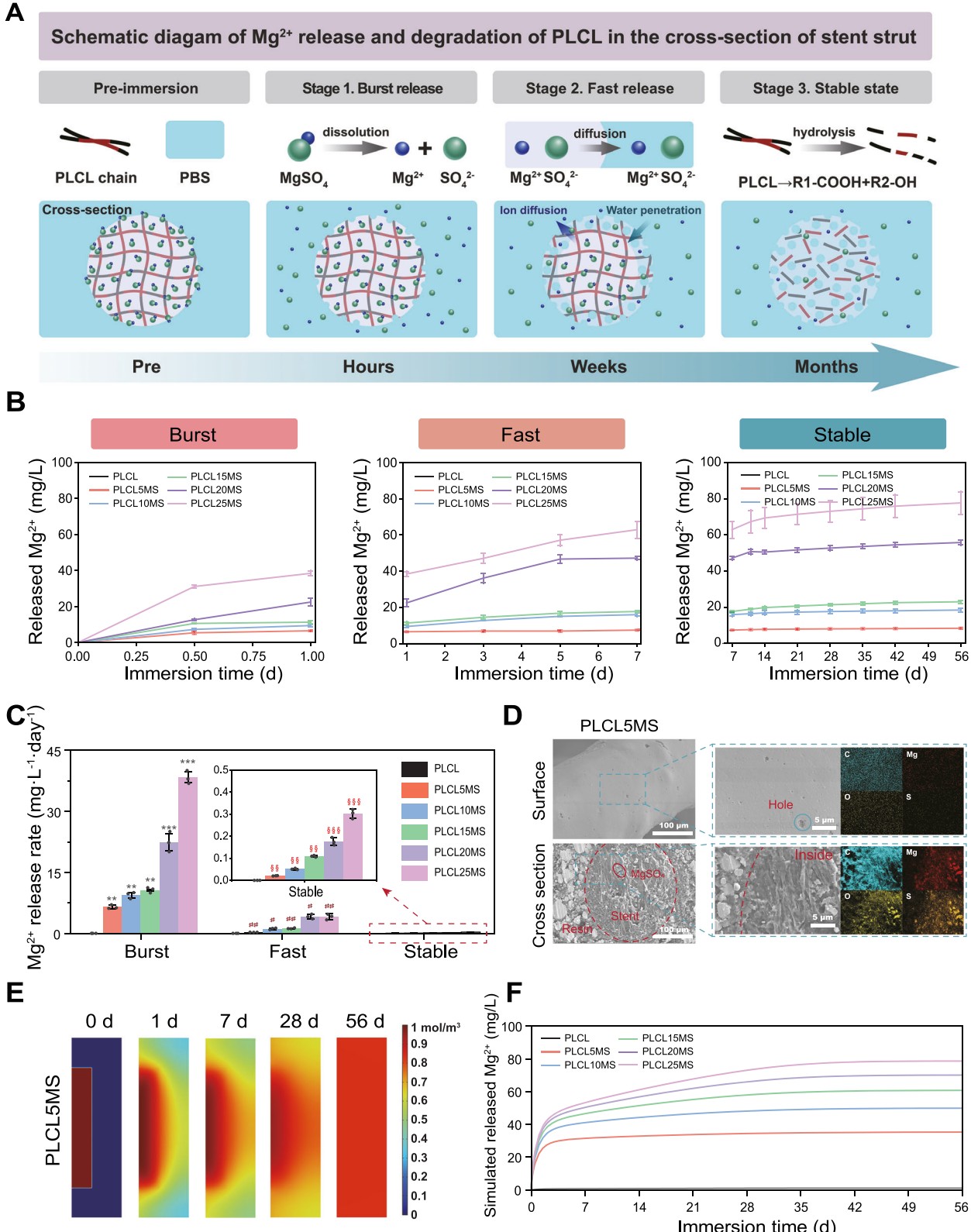

**Fig. 3 | Staged in vitro degradation behavior of the PLCLxMS stents in PBS solution at different time points. A** Schematic diagram of Mg²⁺ release and degradation of PLCLxMS stent struts from cross-sectional perspective. **B** Staged-cumulative release curve of Mg²⁺ at different stages. **C** Staged-Mg²⁺ release rate over 56 days. **D** Surface and cross-sectional morphology of the PLCL10MS stents after immersion in PBS for 7 days. **E** Simulation of Mg²⁺ dissolution and concentration change in PLCL5MS composites. **F** Simulated release curves of Mg²⁺ during immersion of composites with varying MgSO₄ contents, based on diffusion theory. Source data and exact P values are provided as a Source data file. One-way analysis of variance (ANOVA) with a Tukey/Games-Howell post hoc test for multiple comparisons. Sample size: $n = 3$ biologically independent replicates. **$P < 0.01$, ***$P < 0.001$, Burst vs. Fast group; # $P < 0.05$, ## $P < 0.01$, ### $P < 0.001$, Fast vs. Stable group; §§ $P < 0.01$, §§§ $P < 0.001$, Stable vs. Burst group. Data are presented as mean values ± SD.

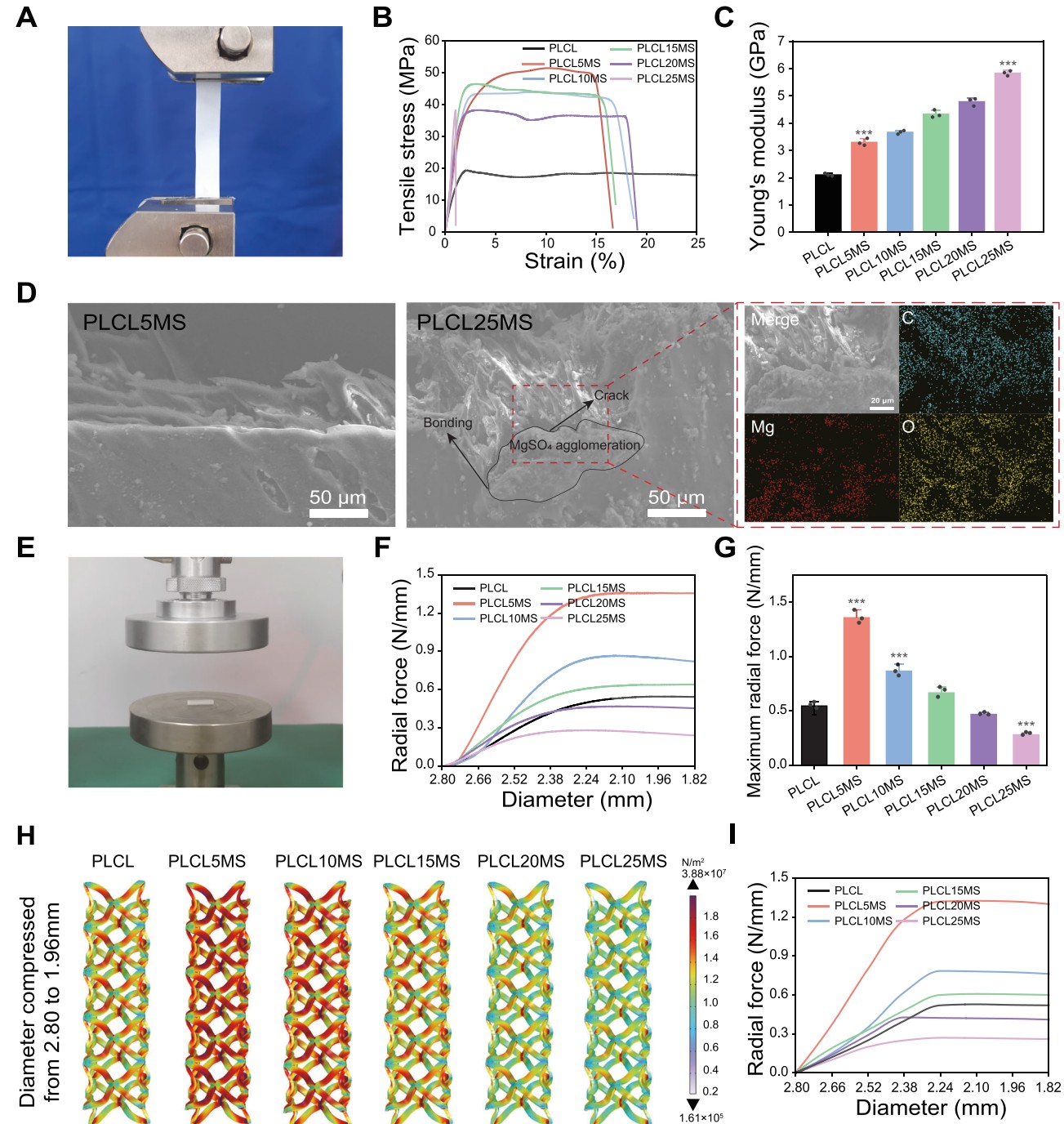

**Fig. 4 | Mechanical properties of PLCLxMS composites and stents. A** Tensile test device for PLCLxMS sheets. **B** Stress-strain curve of PLCLxMS sheets. **C** The calculated Young's modulus of PLCLxMS sheets. **D** SEM images and EDS mapping of the fracture topography of PLCL5MS and PLCL25MS sheets post-tensile test. **E** Radial force test device. **F** Radial force with compression of PLCLxMS stents. **G** The maximum radial force with compression of PLCLxMS stents. **H** Simulated body stress distribution in PLCLxMS stents when compressed to 70% of their original diameter. **I** Simulated radial force with compression of PLCLxMS stents. Source data and exact P values are provided as a Source data file. One-way analysis of variance (ANOVA) with a Tukey/Games-Howell post hoc test for multiple comparisons. Sample size: $n = 3$ biologically independent replicates. ***$P < 0.001$, PLCL vs. PLCL5MS group; PLCL vs. PLCL10MS group; PLCL vs. PLCL25MS group. Data are presented as mean values ± SD.

To further understand the mechanical strengthening mechanism of PLCL by MgSO₄, an exploratory experiment focused on the interactions between $Mg^{2+}$ and the oligomers and monomers produced by PLCL degradation was conducted. First, the FTIR spectra (Fig. 5B) of PLCL5MS stents after immersion in PBS shows that the characteristic peaks at $704\,cm^{-1}$ due to C−H bending did not change until the end of the "Fast" stage[53]. However, a blue shift (from $704\,cm^{-1}$ to $692\,cm^{-1}$) of

the peaks appeared at the "Stable" stage, which was not found in pure PLCL stents (Fig. S8A). Then PLCL sheets were immersed in different solutions to investigate the impact of $Mg^{2+}$ during PLCL degradation (Fig. 5C). The hardness of PLCL before immersion ($21.7 \pm 0.8$) is higher than that after immersion in deionized water ($19.3 \pm 1.3$) but lower than that after immersion in MgSO₄ solution ($23.8 \pm 1.4$), as shown in Fig. 5D. The decrease in hardness can be attributed to the hydrolysis of the

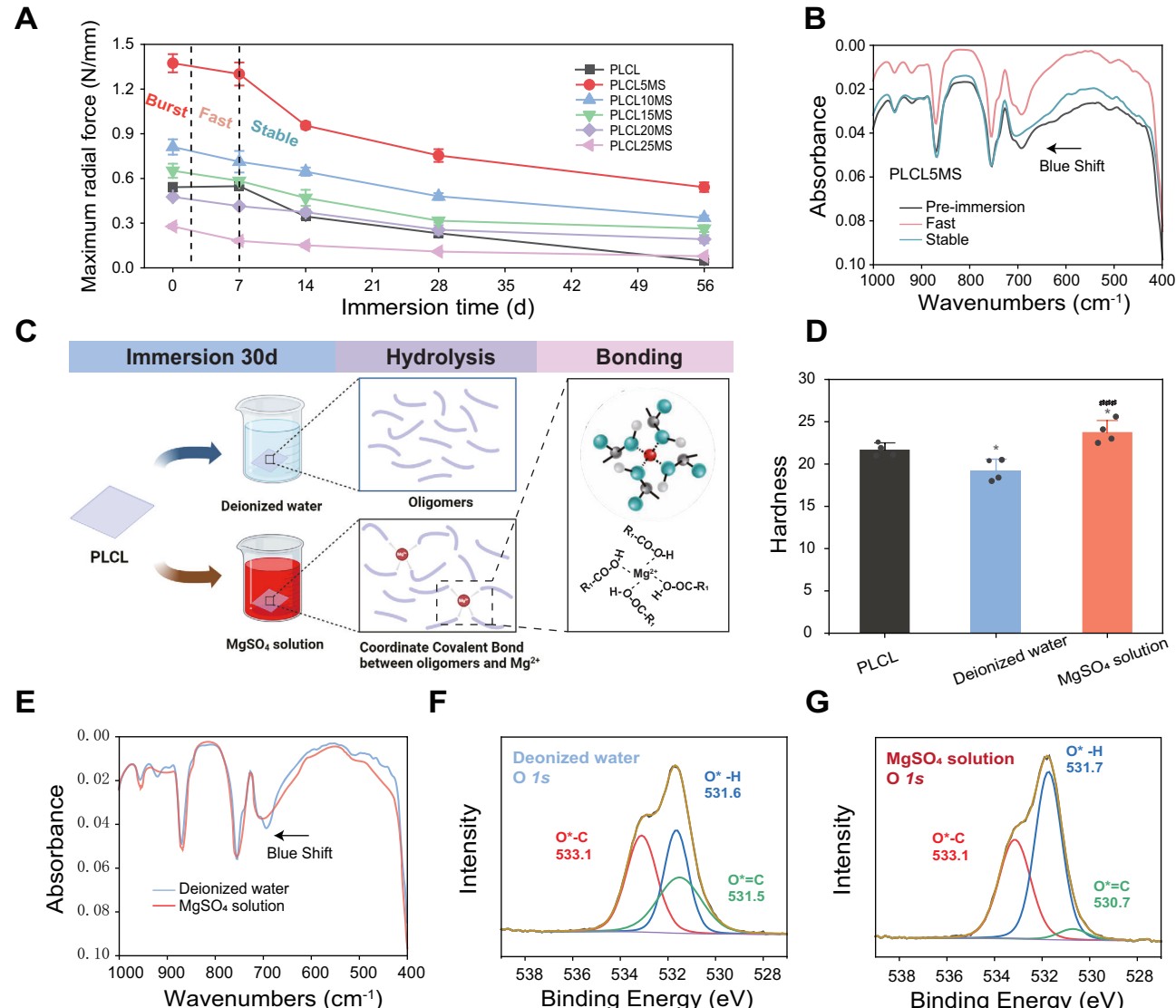

**Fig. 5 | Investigation into mechanical strengthening mechanisms during PLCL degradation. A** Staged-radial force of PLCLxMS stents measured after immersion for 7, 14, 28 and 56 days. **B** FTIR spectra of PLCL5MS stents after immersion for 7 days (Fast) and 56 days (Stable). **C** Schematic diagram illustrating experiments and conjectures to investigate the reinforcement mechanism of $Mg^{2+}$ during PLCL degradation (Created in BioRender[123]). **D** Hardness of the PLCL surface after immersion in various solutions. **E** FTIR spectra of the PLCL surface post-immersion. **F, G** High-resolution *O1s* XPS spectra of PLCL following immersion. Source data and exact P values are provided as a Source data file. One-way analysis of variance (ANOVA) with a Tukey/Games-Howell post hoc test for multiple comparisons. Sample size: $n = 4$ biologically independent replicates. *$P < 0.05$, PLCL vs. Deionized water group or PLCL vs. $MgSO_4$ solution group; ### $P < 0.001$, Deionized water vs. $MgSO_4$ solution group. Data are presented as mean values ± SD.

copolymer, which results in the formation of oligomers and monomers[41]. Conversely, the increase in hardness can be explained by the same blue shift as Fig. 5B observed in Fig. 5E, indicating the formation of new bonds or interactions between the oligomers, monomers, and $Mg^{2+}$. These interactions can alter the properties and behavior of PLCL[54,55]. Additionally, the XPS spectrum (Fig. S8) and the *O1s* high-resolution spectra in Fig. 5F, G also show variations in the intensity of the binding O-H, indicating the presence of a new force formed by the oxygen in the carboxyl group of oligomers and monomers. This suggests the potential for chelation formation between $Mg^{2+}$ and the oxygen element[56].

### In vitro biocompatibility test

For cytotoxicity analysis, HUVECs, HASMCs, and HT-22 cells were selected to simulate toxicity to blood vessels and neurons, respectively. CCK-8 assays indicated that extracts of PLCL loaded with various concentrations of $MgSO_4$ had no significantly cytotoxic effects on

HUVECs, HASMCs, and HT-22 in vitro (Fig. S9A–C). The extracts improved HUVECs proliferation at 1 day (PLCL, PLCL5MS, PLCL10MS) and 7 days (PLCL), and enhanced HASMCs proliferation at 3 and 5 days (PLCL, PLCL5MS, PLCL10MS). Moreover, there was no significant difference in the dead/live cell ratio compared to control groups, according to live/dead cell staining (Fig. S9D). Thus, during degradation, PLCL exhibited no apparent cytotoxicity to vessels or neurons.

The in vitro cell experiments demonstrated that all sample extracts exhibited good biocompatibility with HUVECs, HASMCs, and HT22 cells. The study also found that the pure PLCL sample group had a positive effect on the proliferation of endothelial and smooth muscle cells, although this promoting effect was diminished as the magnesium ion concentration in the solution increased.

According to the platelet adhesion test, no obvious platelet aggregations were observed on the surface of the PLCL, PLCL5MS, and PLCL10MS before and after immersion for 7 days (Fig. S10A), indicating that the platelets were not activated[57]. The hemolysis rates of all

materials were less than 5%, classifying them as blood-compatible materials according to the ISO 10993-4 standard (Fig. S10B). These results suggest that the dissolution of MgSO$_4$ and the surface morphology changes in PLCL5MS and PLCL10MS had minimal effects on their hemocompatibility.

### In vitro neuroprotection analysis using the OGD/R cell model
Given that PLCL loaded with different concentrations of MgSO$_4$ could produce a gradient of Mg$^{2+}$ after immersion (Fig. S11), the neuroprotective effects of different PLCLxMS extracts were further

evaluated using the OGD/R cell model to mimic ischemia/reperfusion injury. According to cell viability assays, after OGD/R treatment, neural viability decreased, which was not mitigated by extracts of pure PLCL but was suppressed by extracts of PLCLxMS (Fig. 6B). The results were very similar for the SH-SY5Y cell line in Fig. S12. Conversely, LDH release, which indicates cell death, increased after OGD/R; extracts of pure PLCL had no effect on this increase. However, extracts of PLCL loaded with MgSO$_4$ could restrain LDH release (Fig. 6C). ROS and Ca$^{2+}$ overload, common causes of cell damage after OGD/R, were also measured. OGD/R injury increased ROS

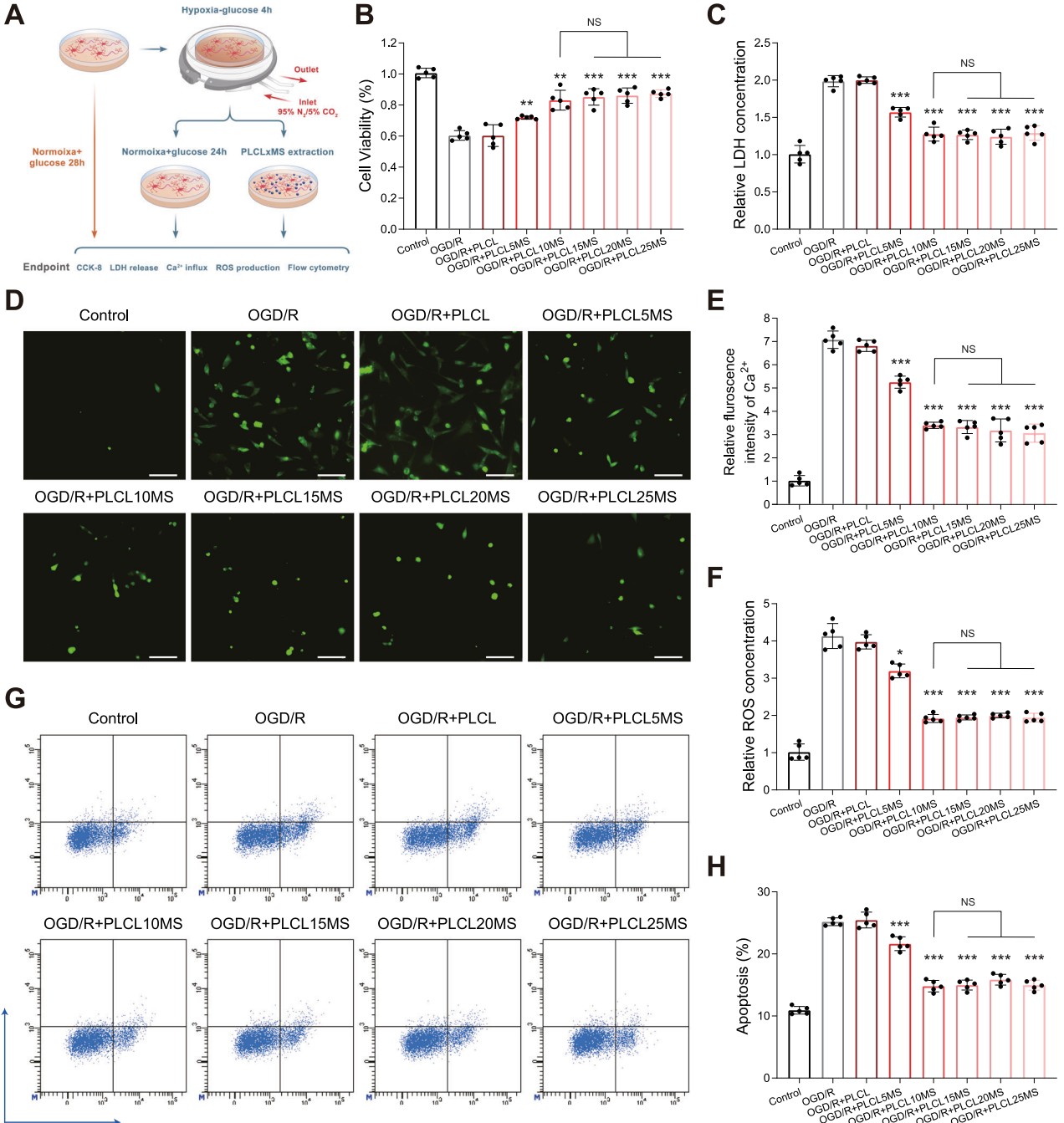

**Fig. 6 | In vitro neuroprotection analysis using the OGD/R cell model.**
**A** Schematic of the cell OGD experiment. **B** Cell viability was assessed by CCK-8 assays. **C** Relative LDH concentration released from the neuron to the supernatant. **D** Typical images of intracellular Ca$^{2+}$ observed by fluorescent microscopy; scale bars: 20 μm. **E** Quantification of relative fluorescence intensity of intracellular Ca$^{2+}$. **F** Relative fluorescence intensity of ROS in neurons. **G** Changes in cell apoptosis

detected by flow cytometry. **H** Quantification of apoptosis rate from (**G**). Source data and exact P values are provided as a Source data file. One-way analysis of variance (ANOVA) with a Tukey/Games-Howell post hoc test for multiple comparisons. Sample size: $n = 5$ biologically independent replicates. *$P < 0.05$, **$P < 0.01$, ***$P < 0.001$ vs. OGD/R group. NS, not significant. Data are presented as mean values ± SD.

production and intracellular $Ca^{2+}$ levels. Although pure PLCL did not reduce intracellular ROS and $Ca^{2+}$, PLCLxMS inhibited these trends (Fig. 6D–F). From the perspective of cell death, the apoptosis rate of neurons was reduced by PLCLxMS but not by pure PLCL (Fig. 4G, H). Among different samples, PLCL10MS and higher $MgSO_4$ contents exhibited more significant effects on viability rescue and inhibition of $Ca^{2+}$ influx, ROS production, and cell apoptosis, though no dose-related effect was observed beyond 10% $MgSO_4$ (Fig. 6B–H). These results suggest that PLCL10MS provided better neuroprotective effects on OGD/R cells than PLCL5MS.

### Ex vivo neuroprotection analysis

Given the superior mechanical properties and significant neuroprotection of PLCL5MS, and the better neuroprotective effect and mechanical properties of PLCL10MS compared to pure PLCL, PLCL5MS and PLCL10MS wires were selected as stent implant alternatives for further in vivo experiments using tMCAO rat models, with pure PLCL wires serving as the control group (Fig. 7A). All in vivo neurological analyses were conducted by implanting PLCLxMS wires into the tMCAO models.

The tMCAO group exhibited significant infarct injury, and pure PLCL wire implantation had no effect on the infarction area. However, the tMCAO+PLCL5MS and tMCAO+PLCL10MS groups could reduce the infarction area, with the tMCAO+PLCL10MS group exhibiting the smallest infarction area (Fig. 7B, C). Moreover, the results of Nissl staining were consistent with triphenyl tetrazolium chloride (TTC) staining. The tMCAO group suffered severe neural damage in the cortex and striatum, which could not be mitigated by pure PLCL wire implantation. In contrast, PLCL5MS and PLCL10MS wire implantations inhibited neural damage, especially in the PLCL10MS group, which displayed more healthy neurons in the striatum than the PLCL5MS group (Fig. 7D, E). Furthermore, the blood-brain barrier (BBB) in the tMCAO group and tMCAO+PLCL group underwent severe damage, leading to pronounced brain edema and Evans blue extravasation. However, the BBB function in the tMCAO+PLCL5MS and tMCAO +PLCL10MS groups was more intact, particularly in the tMCAO +PLCL10MS groups (Fig. 7F–H). Additionally, relative cerebral blood flow (CBF) was measured. On the first day after reperfusion, both the PLCL5MS and PLCL10MS groups showed better CBF recovery than the tMCAO and tMCAO+PLCL groups. Moreover, the PLCL10MS group exhibited the best CBF recovery on day 7 after reperfusion. At the end of each laser speckle imaging experiment, the $pCO_2$ levels and blood pressure of all rats were recorded, with minimal variations (Fig. S13), consistent with observations made previously through direct intra-arterial infusion of $MgSO_4$[58].

Based on the excellent in vivo neuroprotective effect of PLCL10MS, wires were further upgraded to spiral stents (Fig. S14A) and implanted in the CCA after tMCAO (Fig. S14B). The PLCL10MS spiral stents still provided neuroprotection against I/R injury on the first day after implantation (Fig. S14C), further demonstrating the in vivo neuroprotective effects of PLCLxMS, not only in the form of wires but also spiral stents.

After assessing brains integrity, the behavioral test of rats was further evaluated. According to the Longa score, after stroke and reperfusion injury, the tMCAO group exhibited severe neurological disorders, and implantation of pure PLCL wires (tMCAO+PLCL) could not mitigate neural damage. However, rats in the tMCAO+PLCL5MS and tMCAO+PLCL10MS groups displayed better neurological function, with the Longa score of tMCAO+10% being lower than that of tMCAO+PLCL5MS (Fig. 8A). Moreover, rats in the tMCAO+PLCL5MS and tMCAO+PLCL10MS groups spent more time on the rotating rod and less time contacting and removing tapes than those in the tMCAO and tMCAO+PLCL groups. Additionally, the tMCAO+PLCL10MS group displayed the best motor coordination ability (rotarod test) and sensorimotor function (adhesive test) (Fig. 8B–F).

Based on the open field test, which assessed autonomous activity and post-stroke anxiety, rats in the tMCAO+PLCL5MS and tMCAO +PLCL10MS groups were more inclined to actively explore the entire area and stay in the central and internal periphery areas compared to tMCAO and tMCAO+PLCL groups. While there was no significant difference between the distances and time spent in the center by the tMCAO+PLCL5MS and tMCAO+PLCL10MS groups, the tMCAO +PLCL10MS group covered more distance and spent more time in total and surrounding areas than the tMCAO+PLCL5MS group (Fig. 8G–H). All the behavioral test results indicated that rats in the tMCAO +PLCL5MS group exhibited better mobility than those in the tMCAO group from day 1 to day 14, although their performance was not as good as that of the tMCAO+PLCL10MS group. These results suggest that both PLCL5MS and PLCL10MS provided significant neuroprotection, which was evident from day 1 through day 14.

To address the effect of PLCLxMS on the reperfusion injury, the wire was also implanted into a permanent MCAO model (pMCAO) with no reperfusion for comparison and it was found that the neuroprotection of PLCL10MS in pMCAO is less effective than that in tMCAO (Fig. S15), which is consistent with findings in clinical studies.

### Ex vivo $Mg^{2+}$, ROS, and $Ca^{2+}$ concentration detection

As depicted in Fig. 8I, $Mg^{2+}$ concentrations in the blood of rats from the experimental group (tMCAO+ PLCL10MS and tMCAO+PLCL5MS) were higher than those in the control group (sham, tMCAO, and tMCAO +PLCL) on days 1 and 7. $Mg^{2+}$ concentrations in the ipsilateral infarction brain (IIB) of rats in the experimental group were also higher on day 1 but showed no differences by day 7. Moreover, $Mg^{2+}$ levels in the infarction contralateral brain (ICB) at 1 and 7 days post-implantation of PLCLxMS wires remained similar to those of the control groups (Fig. S16). Endothelialization on the implanted wires was not observed by day 7 (Fig. S17). One possible explanation is that the recovery of BBB integrity in tMCAO rats by day 7 facilitated the dynamic equilibrium of $Mg^{2+}$ concentrations in the brain[59].

In addition to $Mg^{2+}$, cerebral ROS and $Ca^{2+}$ concentrations were also measured on day 1 (Fig. S18). Consistent with in vitro findings, ROS and $Ca^{2+}$ levels in the ICB did not differ among the groups. However, in the IIB, ROS and $Ca^{2+}$ levels were elevated in the tMCAO and tMCAO +PLCL groups, while the implantation of PLCL5MS and PLCL10MS wires significantly inhibited this increase (Fig. S18A–B), with the tMCAO+PLCL10MS group showing the most effective inhibition.

### Implantation result and histological analysis of PLCL10MS stent in rabbit CCA model

The optical morphology of the PLCL and PLCL10MS stents during crimp and expansion are shown in Fig. S19. Digital Subtraction Angiography images (DSA) in Fig. S20 show that both of the PLCL and the PLCL10MS stents were successfully implanted and no stenosis were observed at 7 days and 1 month. Combined with the SEM (Fig. S21) and HE staining (Fig. S23) results, the PLCL and PLCL10MS stents struts in rabbits CCA were partly covered by the neointima at 7 days and were fully embedded by intima at 1 month. The concentration of magnesium at 1 day in CCA blood of PLCL10MS stent group was higher than that of the sham and PLCL stent group, but with no statistical difference at 7 day (Fig. S23).

## Discussion

This study focused on the preparation and characterization of PLCLxMS composite stents for treating cerebral large artery atherosclerosis. Different concentrations of $MgSO_4$ were incorporated into the stents to provide neuroprotection for damaged brain tissues after acute stroke and to enhance their inherent mechanical properties. The mechanical strengthening mechanisms of $MgSO_4$ particles and $Mg^{2+}$ ions, post-degradation at the micron and nano scales, were thoroughly analyzed. Additionally, the release of magnesium ions and the

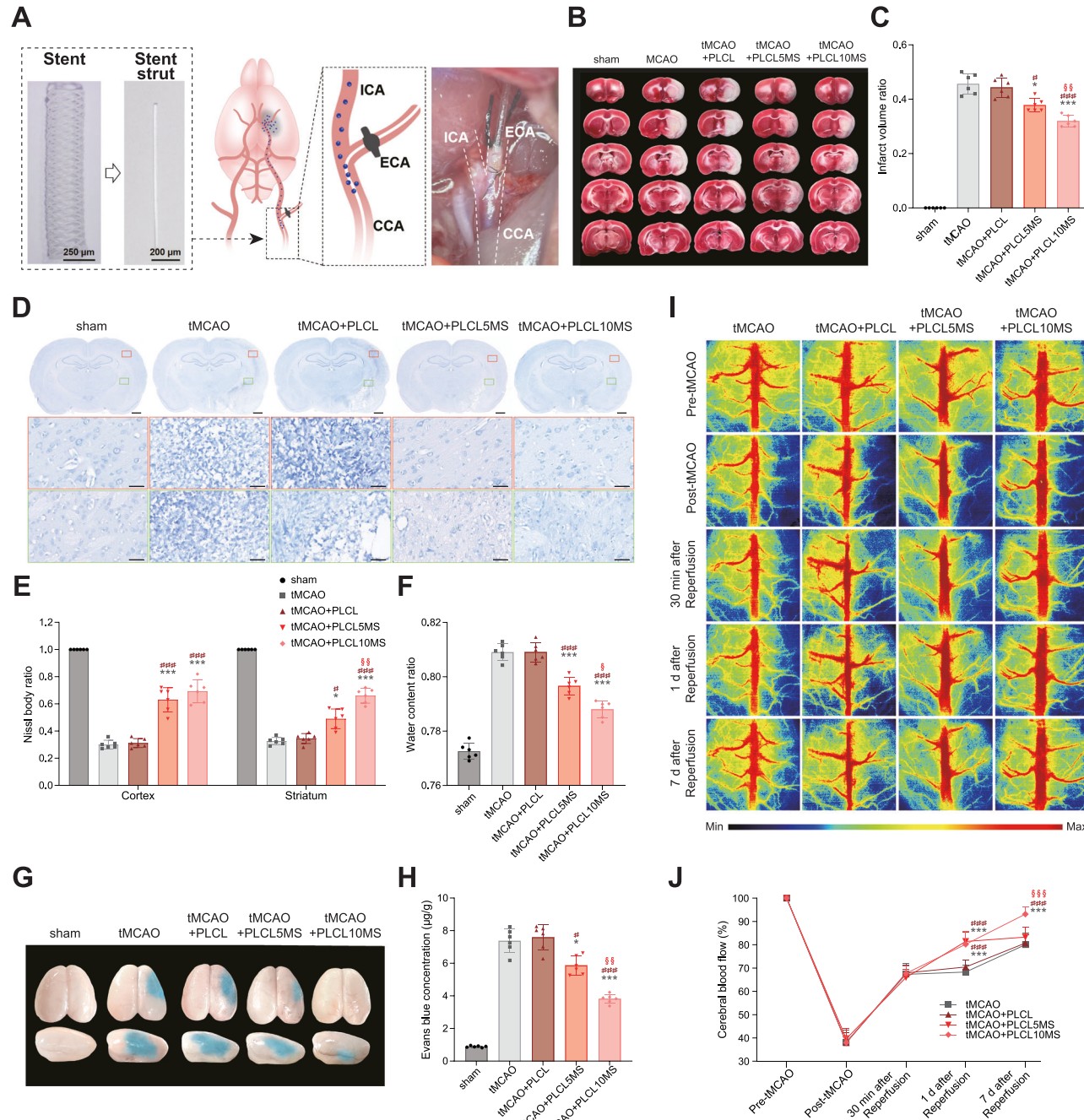

**Fig. 7 | In vivo neuroprotection analysis of PLCLxMS in tMCAO rat models on the 7 days after ischemia/reperfusion injury. A** Schematic diagram of stent strut implantation process. **B** Brain slices stained with TTC to differentiate between healthy tissue (red) and damaged tissue (white). **C** Quantification of brain infarct volume from (**B**). **D, E** Brain slices with Nissl staining and quantification of Nissl bodies; scale bars: 1 mm and 25 μm. **F** Evaluation of brain water content ratio. **G** Representative Evans blue extravasation image illustrating blood-brain barrier disruption. **H** Quantification of cerebral Evans blue content. **I** Measurements of CBF before, during, and after surgery. **J** Comparison of relative CBF across groups. Source data and exact P values are provided as a Source data file. Two-way analysis of variance (ANOVA) with a Tukey/Games-Howell post hoc test for multiple comparisons. Sample size: $n = 6$ biologically independent replicates. *$P < 0.05$, **$P < 0.01$, ***$P < 0.001$, tMCAO+PLCL5MS or tMCAO+PLCL10MS vs. tMCAO group; # $P < 0.05$, ## $P < 0.01$, ### $P < 0.001$, c or tMCAO+PLCL10MS vs. tMCAO+PLCL group; §$P < 0.05$, §§ $P < 0.01$, §§§ $P < 0.001$, tMCAO+PLCL10MS vs. tMCAO+PLCL5MS. Data are presented as mean values ± SD.

mechanisms underlying their neuroprotective effects were investigated both in vitro and in vivo.

Stent placement for treating cerebrovascular disease is an emerging technique. Unlike coronary arteries, it is challenging to fully conform stents to the anatomical curvature of cerebral vessels, leading to potential complications such as cerebral vessel perforation, entrapment, and spasm. Intracranial stenting was not introduced until 1998, and in 2004, the U.S. FDA approved percutaneous transluminal angioplasty and stenting for treating ischemic cerebrovascular diseases. The SAMMPRIS and VISSIT randomized controlled trials (RCTs) demonstrated that intracranial stenting was less effective than drug therapy, primarily due to high complication rates associated with stenting[60]. Additionally, combining stenting with medical therapy did not significantly reduce the risks of stroke or death in patients with symptomatic intracranial stenosis compared to medical therapy alone[61]. A potential explanation for these suboptimal stenting

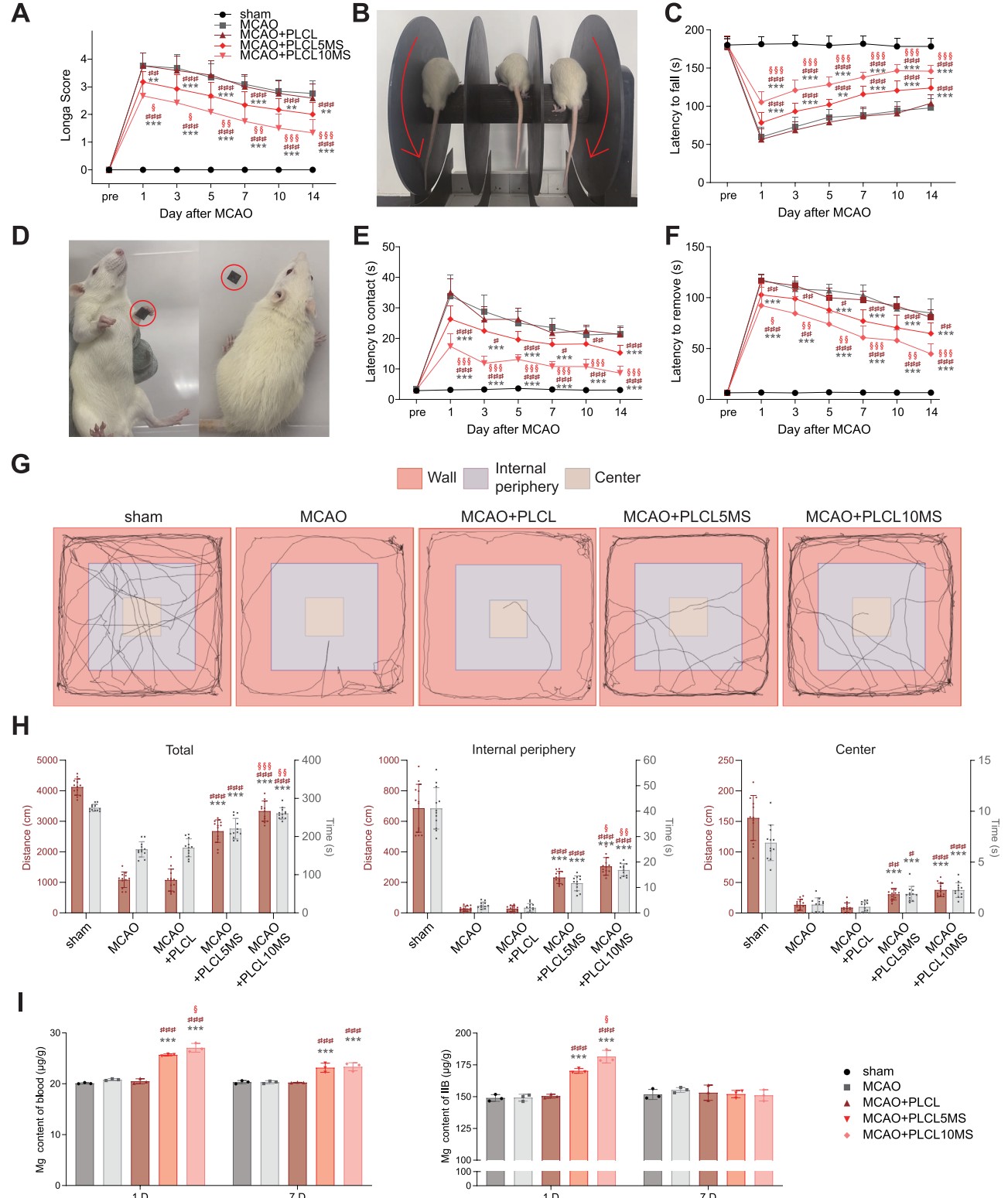

**Fig. 8 | Behavioral evaluation of tMCAO rat models implanted with PLCLxMS wires after ischemia/reperfusion injury. A** Longa score assessment. **B, C** Results from the Rotarod test. **D**–**F** Adhesive contact and removal test performance before stroke and up to 14 days post-stroke. **G** Open field test conducted 7 days post-stroke. **H** Walking distance and time in the open field. **I** Magnesium concentration in the blood and ipsilateral infarction brain (IIB) of rats at 1 and 7 days post-implantation of PLCLxMS wires. Source data and exact P values are provided as a Source data file. Two-way analysis of variance (ANOVA) with a Tukey/Games-Howell post hoc test for multiple comparisons. Sample size: *n* = 12 for neurological tests, biologically independent replicates; *n* = 3 for magnesium content measurement tests, biologically independent replicates. ***P* < 0.01, ****P* < 0.001, tMCAO +PLCL5MS or tMCAO+PLCL10MS vs. tMCAO group; # *P* < 0.05, ## *P* < 0.01, ### *P* < 0.001, tMCAO+PLCL5MS or tMCAO+PLCL10MS vs. tMCAO+PLCL group; § *P* < 0.05, §§ *P* < 0.01, §§§ *P* < 0.001, tMCAO+PLCL10MS vs. tMCAO+PLCL5MS. Data are presented as mean values ± SD.

outcomes could relate to cerebral ischemia-reperfusion injury or cerebral hyperperfusion syndrome[62]. Recanalizing occluded vessels and restoring blood flow in ischemic strokes often trigger various molecular and cellular responses, underscoring the importance of neuroprotective strategies post-recanalization

Current biodegradable materials for medical applications primarily encompass biodegradable metals and polymers. Among these, biodegradable polymers demonstrate superior suitability for cerebrovascular stents due to their enhanced flexibility and malleability, which facilitate better adaptation to the intricate anatomy of intracranial vasculature. As summarized in Table S2[63–67], poly-L-lactic acid (PLLA) represents the most widely utilized biodegradable polymer. Notably, the Igaki-Tamai stent, recognized as the first self-expanding PLLA-based resorbable stent implanted in humans, marked a significant milestone in this field[68]. However, PLLA exhibits certain limitations, including inherent brittleness and relatively slow degradation kinetics[68]. In contrast, poly(L-lactide-co-ε-caprolactone) (PLCL), a copolymer synthesized from lactic acid (LA) and ε-caprolactone (CL), has emerged as an optimized alternative to PLLA. This advanced material offers several distinct advantages: (1) accelerated degradation rates[53], making it particularly suitable for applications requiring short-term support (typically 3-6 months); (2) enhanced flexibility and superior elongation at fracture; and (3) improved processing characteristics[69]. These properties not only address the limitations of PLLA but also render PLCL particularly advantageous for advanced manufacturing techniques, including 3D printing applications.

The 3D printing methodology employed in this study demonstrates both cost-effectiveness and manufacturing flexibility[69]. This technology has gained increasing recognition as a promising approach for fabricating personalized biodegradable polymer stents[70], particularly for small-batch production and highly customized applications. The PLCLxMS stents were fabricated using fused deposition modeling (FDM), a process wherein a heated nozzle precisely extrudes molten thermoplastic material along predetermined paths, with subsequent solidification forming the final product. Compared to conventional laser cutting techniques, FDM offers several advantages: (1) reduced material consumption, (2) lower manufacturing costs, and (3) enhanced capability for producing complex, patient-specific stent geometries, thereby ensuring the practical feasibility of PLCLxMS stent production. While 3D printing of pure PLCL stents has been established as a low-cost process[69], the fabrication of PLCLxMS stents utilizes identical equipment and processing parameters, with the modification of incorporating a PLCL-magnesium sulfate composite. Consequently, although the production cost of PLCLxMS stents is marginally higher than that of pure PLCL stents, it remains within a cost-effective range for medical device manufacturing.

Ischemic stroke is a dynamic process of infarct expansion influenced by time, residual blood flow, and other factors[71]. The traditional time window for acute ischemic stroke intra-arterial therapy (IAT) is less than 6 h, based on pharmacological thrombolysis without penumbral imaging[72]. During cerebral ischemia, recanalizing occluded vessels and restoring blood flow often triggers molecular and cellular responses, including $Ca^{2+}$ overload and ROS production, which are key factors in the development of reperfusion neural injury.

After the implantation of PLCLxMS stents, the $Mg^{2+}$ generated by the dissolution of $MgSO_4$ particles transferred to the distant ischemic area and cross the BBB into the brain, contributing to neuron rescue, BBB protection, and enhanced CBF. Other reports suggested that injecting magnesium directly into target organs via proximal arteries might induce stronger neuroprotection[26]. A supplementary experiment was conducted to compare the neuroprotective effects of PLCL10MS implantation and intravenous injection of $Mg^{2+}$. It was found that these two treatment approaches had similar neuroprotective trends, but PLCL10MS exhibited superior neuroprotective effects (Fig. S23). Magnesium ion supplementation provides neuroprotective effects through several mechanisms. Magnesium acts as an endogenous antagonist of voltage-and ligand-gated calcium channels and the glutamate-N-methyl-D-aspartate (NMDA) receptor, which can reduce excitotoxicity caused by calcium overload after nervous system injury and decreases brain cell death[20,24,73]. It also dilates cerebral vessels, enhancing blood flow post-cerebral ischemia, stabilizes mitochondrial function, and regulates enzymes critical for energy metabolism in the brain post-ischemia[74,75]. In this study, the HT22 cell death and apoptosis after glucose-oxygen deprivation were notably reduced in the extract of PLCLxMS composites. This outcome aligns with subsequent experiments showing a decrease in intracellular calcium ion concentration and a reduction in intracellularly released LDH, corroborating the neuroprotective function of the extracts both in vitro and in vivo.

Magnesium neuroprotection in acute stroke in clinical settings is controversial, possibly due to the timing of administration[76]. Recent studies have suggested that neuroprotectants should be tested in the reperfusion era, coinciding with the widespread application of mechanical thrombectomy[77]. The prehospital administration of $MgSO_4$ was deemed safe and enabled treatment initiation within two hours after stroke onset. However, it failed to show any improvement in clinical outcomes over a 90-day period in a randomized clinical trial[78]. One potential reason is that patients with AIS did not undergo thrombectomy or thrombolytic therapy; without initial reperfusion, the effectiveness of neuroprotective agents alone may be limited[79]. For comparison, the wire was also implanted into a permanent MCAO model (pMCAO) with no reperfusion, and it was found that the neuroprotection provided by PLCL10MS in pMCAO was less effective than in tMCAO (Fig. S15), aligning with clinical study findings[78]. Consequently, reperfusion therapy (stent implantation) was combined with neuroprotective treatment ($Mg^{2+}$ administration) by incorporating $MgSO_4$ particles into a PLCL stent to mitigate cerebral ischemia-reperfusion injury.

In this study, soluble $MgSO_4$ as a filler enables the release of $Mg^{2+}$ to produce a staged effect. Bo Chen et al. established magnesium oxide-reinforced 3D-printed composite stents and found that the accumulative Mg concentrations of $Mg^{2+}$ released in simulated body fluid (SBF) from 1 to 30 days increased linearly[32]. However, this study observed that the disappearance of $MgSO_4$ particles indicated complete dissolution and release into solution by day 7. The cumulative release curve of $Mg^{2+}$ could be divided into three stages. Initially, within hours, the concentration of $Mg^{2+}$ increases rapidly due to the dissolution of $MgSO_4$ embedded in the surface. Then, before day 7, the concentration of $Mg^{2+}$ continues to increase but at a reduced rate compared to the first stage. A similar trend of decreased speed also appeared in the pH value and remaining mass of the stent in the third stage.

The phenomenon of phased release of $Mg^{2+}$ is determined by the solubility of $MgSO_4$ and the permeation of water in PLCL. Initially, the dissolution of surface $MgSO_4$ predominates the ion release behavior. Subsequently, the dissolution of $MgSO_4$ inside the stents dominates the release process, dependent on the diffusion velocity of water inside degradable polymers, which can be described by diffusion theory. The time, $t_{diff}$, water needs to travel a mean distance $<x>$ can be estimated by applying random theory to the motion of water in the polymer[41]:

$$t_{diff} = <x> \frac{2\pi}{4D_{eff}} \tag{1}$$

where $D_{eff}$ is the effective diffusion coefficient of water inside PLCL. This equation fits well in the degradation simulation section with the release of $Mg^{2+}$ in the second and third stages, assuming $MgSO_4$ particles are uniformly distributed within the stent.

The neuroprotective effect of $Mg^{2+}$ is dose-related[80]. M. B. Marinov et al. found that animals with middle cerebral artery (MCA)

## Time-Match Between Mg²⁺ Release and Neuroprotection

**Fig. 9 | Illustration of the time-matched release of Mg²⁺ with the treatment windows for AIS.** Stage 1: The "Burst" stage of Mg²⁺ release corresponds to the hyperacute stage of AIS characterized by severe neural injury. The stent exhibits a substantial initial burst release of Mg²⁺, which provides high-dose neuroprotective effects by rapidly diffusing across the compromised BBB to target ischemic death neurons. Stage 2: The "Fast" stage of Mg²⁺ release corresponds to the acute stage of AIS characterized by cerebral edema formation. Mg²⁺ is released at a controlled yet therapeutically effective rate—reduced from the "Burst" stage but maintained at levels sufficient to support neural repair mechanisms including BBB and neuronal repair. The presence of Mg²⁺ attenuates secondary injury while facilitating progressive neuronal recovery. Stage 3: The "Stable" stage of Mg²⁺ release corresponds to the subacute stage of AIS characterized by active brain tissue remodeling. Mg²⁺ release kinetics transition to a slow, sustained profile, ultimately achieving dynamic equilibrium. BBB integrity is restored, and neuronal recovery advances. The radial force of the stent is significantly reduced.

occlusion given 90 mg/kg MgSO₄ prior to 1.5 h of ischemia showed a more significant reduction in infarct volume compared to the 30 mg/kg group[25]. This dose-related phenomenon is also confirmed in this study, where the PLCL10MS group showed more reduction in infarct volume, performed better in behavioral tests, and detected less Ca²⁺ and ROS release compared with the PLCL5MS group. However, it is not feasible to indefinitely increase the MgSO₄ content in the design of composite materials without compromising other properties, such as mechanical integrity. Therefore, it is crucial to regulate the release of Mg²⁺ to ensure they provide optimal neuroprotection within their limited loading capacity. One limitation is that the quantification of ROS levels and Mg²⁺ distribution was analyzed ex vivo, rather than using an in vivo approach. With the development of molecular probe technology, in situ evaluation of reactive species in the rat brain should be conducted.

In this study, the proposed time match between the release of Mg²⁺ and the treatment windows for ischemic stroke were illustrated in Fig. 9 and Table S3[17,20,24,73–75,81–83]. During the "Burst" stage, substantial

amounts of Mg²⁺ ions are released, exerting neuroprotective effects in the hyperacute stage of AIS treatment. The subsequent "Fast" stage features sustained Mg²⁺ release, which facilitates neural repair and attenuates inflammatory responses, ultimately reaching dynamic equilibrium in the "Stable" stage.

The hyperacute stage, representing the optimal therapeutic window for AIS[17], is characterized by ROS-mediated neuronal apoptosis and DNA damage. Excitotoxicity occurs through Ca²⁺ accumulation and glutamate-induced overstimulation of NMDA receptors. These primary injuries are compounded by ROS-specific inflammatory responses, leading to secondary damage. The released Mg²⁺ ions function as endogenous antagonists of voltage- and ligand-gated calcium channels, including NMDA receptors, thereby reducing ROS production and mitochondrial damage. This mechanism effectively suppresses ischemia-reperfusion injury, as demonstrated in Fig. 7. During the acute stage, while inflammatory responses peak, the rate of cerebral infarction volume expansion shows significant reduction[81]. Brain-resident microglia serve as the primary responders to injury,

releasing pro-inflammatory cytokines that disrupt the BBB and exacerbate inflammation[82]. This initial response is subsequently accompanied by ROS-mediated inflammatory cascades[83]. Concurrently, astrocytes proliferate to establish a protective barrier. The $Mg^{2+}$ released during the "Fast" stage inhibits mitochondrial ROS production and regulates hippocampal neural stem cell proliferation[84], thereby promoting neural repair in rats. During the subacute stage, which corresponds to the Stable phase, most reperfusion-induced neural damage is repaired. Meanwhile, magnesium sulfate particles retained within the stent struts maintain the mechanical integrity of the stent through their slow dissolution process. Excitingly, PLCLxMS stents also exhibited better mechanical performance than pure PLCL stents, as shown in Fig. 5A.

In AIS treatment, the staged release of $Mg^{2+}$ should align with the therapeutic time window, given the time-sensitive nature of stroke and the dose-dependent neuroprotective properties of $Mg^{2+}$ [80]. Ghozy et al. demonstrated that ultra-early administration of neuroprotective agents is essential for achieving therapeutic efficacy[85]. This is supported by Marinov et al.'s findings, which revealed that MCAO animal models treated with 90 mg/kg $MgSO_4$ showed significantly greater infarct volume reduction compared to the 30 mg/kg group[25]. These findings suggest that optimizing $Mg^{2+}$ release during the hyperacute stage could potentially enhance the neuroprotective efficacy of the PLCLxMS stent. Beyond the hyperacute stage, timely $Mg^{2+}$ delivery plays a crucial role in mitigating apoptosis and necrosis, preserving neurological function, and promoting the recovery of compromised brain tissue. Furthermore, maintaining optimal stent mechanical integrity throughout the subacute stage is critical for minimizing stent restenosis rates and ensuring long-term therapeutic outcomes. One limitation is that the quantification of ROS levels and $Mg^{2+}$ distribution was analyzed ex vivo, rather than using an in vivo approach. With the development of molecular probe technology, in situ evaluation of reactive species in the rat brain should be conducted.

The mechanical properties of the designed PLCLxMS stents are superior to those of pure PLCL, due to the reinforcement effect of Mg-based inorganic fillers. However, unlike traditional insoluble fillers such as MgO and $MgCO_3$, $MgSO_4$ is a water-soluble salt. Therefore, during the swelling or hydrolysis of polymers, the internal $MgSO_4$ particles quickly release $Mg^{2+}$, which can form chelation interactions with the polymer degradation products, playing a key role in slowing down matrix degradation and enhancing substrate mechanical performance. Thus, we propose a combined mechanical enhancement mechanism involving $MgSO_4$-particle/$Mg^{2+}$-ions.

The application of degradable polymer stents for intracranial atherosclerosis is often limited by insufficient mechanical properties. For instance, the Absorb BVS was withdrawn from the market after nearly 100,000 patients with CAD had been enrolled in multi-center clinical trials or had received the stent implantation globally[63]. A significant reason for its withdrawal was the poor radial strength of PLLA, resulting in a thicker stent wall that increased the probability of restenosis clinically. Addressing this issue, recent decades have seen substantial progress in enhancing the mechanical properties of biomedical polymers. A commonly employed method is the incorporation of fillers such as MgO, ZnO, and carbon nanotubes to develop composites[86–88], where the elastic modulus and compressive strength increase with the addition of inorganic stiff filler[50].

In this research, the $MgSO_4$ filler led to an increase in tensile modulus with increasing filler content. However, compressive strength decreased with an increase in filler content. The radial support strengths of the composites increased due to the particle strengthening effect of magnesium sulfate. The compressive strength of PLCL5MS increased by about 170% compared to pure PLCL. However, nanofillers tend to aggregate when the filler content is too high, due to their large surface areas, as confirmed by the typical morphology of PLCL20MS and PLCL25MS. The larger $MgSO_4$ filler aggregates result in poor interfacial bonding between the fillers and PLCL, leading to ineffective strain transfer across the matrix-filler interface during mechanical loading. Consequently, the radial support strengths decrease with increasing $MgSO_4$ content. Interestingly, the PLCLxMS composites increase the tensile moduli with increasing $MgSO_4$ content but markedly reduce the fracture strain. This decrease also exists in many other fillers, such as hydroxyapatite[89]. This suggests that the PLCL25MS composite is very brittle and unsuitable for making cerebrovascular stents. The combined consideration of radial support and elongation at break indicates that PLCL5MS and PLCL10MS exhibit very good mechanical performance, showing promise for their application in cerebrovascular stents in the future.

The magnesium sulfate used in this study not only acts as a filler impacting the mechanical properties of PLCL, but the released $Mg^{2+}$ also affects the degradation and mechanical properties of PLCL. Research reveals that the hydrolysis process of PLCL is a block hydrolysis. The rate of hydrolysis is influenced by several factors, including pH, type of bond, and molecular weight. In the initial phase, an aqueous solution penetrates the polymer, followed by hydrolytic degradation, which converts the long polymer chains into shorter water-soluble fragments, akin to a reverse polycondensation process. The dissolution of $MgSO_4$, resulting from water penetration, leads to a decrease in solution pH due to the hydrolysis of magnesium ions. This process facilitates the hydrolysis of ester bonds and promotes the degradation of PLCL. The increased degradation rate, as depicted in Fig. 4, shortens the otherwise prolonged degradation time of the stent, which is beneficial for reducing restenosis rates after long-term implantation.

Interestingly, the increased degradation rate did not cause the stent to lose mechanical integrity prematurely during degradation (Fig. 4). Conversely, the composite stent overcomes the brittleness typically associated with PLCL stent degradation over time. This may be attributed to the mechanical enhancement from the $MgSO_4$ particles inside the composite stents that haven't dissolved yet (Fig. 4b). A hypothesis proposed and verified in our study suggests that the maintenance of mechanical strength may be attributed to the formation of coordinate covalent bonds between oligomers, monomers, and $Mg^{2+}$ (Fig. 5).

Previous research on $Mg^{2+}$ neuroprotection primarily focuses on the effects of administration route (intraperitoneal, intravenous, arterial, or intracranial supplementation), administration timing, and $Mg^{2+}$ dose in different stroke models (focal cerebral or global cerebral ischemia)[80]. The objectives of studies on biodegradable Mg-alloy stents are to improve mechanical performance, endothelialization rate, and control substrate corrosion rate through alloying and surface coating (summarized in Table S4)[90–100]. However, in our proof-of-concept study, a novel $Mg^{2+}$-eluting biodegradable neuroprotective stent was proposed. The staged release of $Mg^{2+}$ was designed to provide sequential neuroprotective effects aligned with the treatment window for AIS, verified using a tMCAO rat model to replicate human brain ischemia/reperfusion injury. Moreover, the radial strength of the composite stent was increased compared to the pure PLCL stent, and a novel $MgSO_4$-particle/Mg-ions combined mechanical reinforcement mechanism was introduced to explain this increase. This study may inform the design of novel biodegradable stents for brain disease treatment.

The safety profile of PLCLxMS stents was evaluated through in vitro and in vivo biocompatibility assessments in this study. Initial in vitro investigations demonstrated well hemocompatibility and cytocompatibility. Subsequent one-month implantation studies in both rat and rabbit models revealed complete endothelialization and appropriate inflammatory responses of the stent. However, long-term safety considerations and potential side effects related to stent degradation warrant further investigation. The degradation

mechanism of PLCLxMS stents involves two concurrent processes: magnesium sulfate dissolution and PLCL hydrolysis, both of which contribute to the maintenance of long-term safety[101]. PLCL degradation yields lactic acid and caprolactone, which undergo metabolic conversion to water and carbon dioxide, subsequently eliminated through respiration and urinary excretion, demonstrating excellent long-term biocompatibility[102,103]. Supporting this, Yuval Ramot et al. conducted comprehensive 52-week preclinical studies that validated the long-term biocompatibility and biodegradability of 70:30 PLCL[103]. While the degradation process may potentially induce mild inflammatory responses, foreign body reactions, or localized fibrosis due to lactic acid and caprolactone accumulation[104], the blood-brain barrier's functional integrity effectively regulates lactate and caprolactone diffusion[105]. This regulatory mechanism suggests that PLCL maintains favorable biocompatibility with cerebral tissue throughout its degradation period.

One limitation of our research is that the in vivo neuroprotection evaluation of PLCLxMS was conducted using wires, rather than stents. In reference to the design of biodegradable metal and polymer stents, the safety analysis of these materials typically begins with wires[106–108] and progresses to stents[97,109,110]. For in vivo neurological analysis, the rodent tMCAO model is widely used for ischemic stroke, but it is not suitable for stent implantation[111]. The anatomical complexity of large stroke animal models (swine, sheep, dog, and rabbit) reduces the reproducibility of infarcts, and neurological function assessment methods are not as widely applied in these models as in rodent models[112].

Given the clinical heterogeneity of ischemic stroke subtypes, future investigations should validate the neuroprotective efficacy of PLCLxMS stents across various stroke models, including large vessel occlusion, cardioembolic stroke, and small vessel disease[113]. Although our study has demonstrated the stent's safety and efficacy over a one-month period, several critical aspects require further investigation: (1) the temporal changes in mechanical integrity, (2) long-term in vivo degradation kinetics, and (3) the physiological impact of degradation products. These parameters remain to be fully elucidated and warrant comprehensive evaluation in future studies.

## Methods

We have complied with all relevant ethical regulations declared in the manuscript, and disclosed the name(s) of the board and institution in the methods part. Figs. 1, 3A, 6A, 7A, 9 were created by the authors using Adobe Photoshop (v19.1.9) and are licensed under CC BY 4.0. All experimental procedures received approval from the Capital Medical University's Institutional Animal Investigation Committee and were conducted in accordance with the National Institutes of Health's standards for the Care and Use of Laboratory Animals.

### Stent design and manufacture

Poly (L-lactide-co-caprolactone) copolymer (85% polylactic acid, 15% polycaprolactone by weight%, purchased from Corbion Purac) and magnesium sulfate (≥99%, Reagent grade, Beyotime) were utilized in this study. Magnesium sulfate powder was mixed into the molten PLCL to create a PLCL/MgSO$_4$ mixture with varying MgSO$_4$ contents (denoted as PLCLxMS). Five material formulations, namely PLCL5MS (95 wt.% PLCL, 5 wt.% MgSO$_4$), PLCL10MS (90 wt.% PLCL, 10 wt.% MgSO$_4$), PLCL15MS (85 wt.% PLCL, 15 wt.% MgSO$_4$), PLCL20MS (80 wt.% PLCL, 20 wt.% MgSO$_4$), and PLCL25MS (75 wt.% PLCL, 25 wt.% MgSO$_4$), were fabricated into stents, wires, and sheets using a 3D 4-axial extrusion deposition printer (provided by Beijing Advanced Medical Technologies, Ltd. Inc.)[114]. Briefly, the 3D printing system incorporates a material delivery system that deposits polymer material in the form of hot melt filaments. These filaments adhere to the surface of a rotating rod or to previously extruded filaments already attached to the rod. Stent thickness can be adjusted by varying the speed of the rotation axis to which the polymer filaments are attached or by altering

## Table 1 | Detailed design information of the stent

| Diameter × Length | 2.75 mm×18 mm |
|---|---|
| Number of strut rings | 43 |
| Number of crowns per ring | 7 |
| Number of ring connectors | 172 |
| Strut thickness (Round) | 0.14 mm±0.04 mm |
| Mass per stent | 7.2 ~ 8.3 mg |
| Molecular weight (M$_n$) | 100000 ~ 300000 |

the speed of XY axis movement, similar to the process of hot melt stretching. Stent structures were designed in CAD form, and the CAD data were converted to G-Code commands to guide the printer in depositing thin fibers around the rotating rod to form the stents (Fig. 1). The printing parameters for the stents are detailed in Table 1.

### Morphology and microstructure characterization

The surface topography of the PLCLxMS stents was examined using a scanning electron microscope (SEM) with an Energy Dispersive Spectrometer (EDS, S-4800, Hitachi, Japan) at an accelerating voltage of 10 kV. Fourier-transform infrared (FTIR) spectra of the various stents were recorded using an FTIR spectrometer (Frontier, Perkin-Elmer, USA) in the wavenumber range of 400 ~ 4000 cm$^{-1}$ at room temperature to analyze the chemical groups. Thermal properties of the PLCLxMS composites were determined using DTA (Q600 TGA-DSC-DTA, USA) equipment. Sample compositions were analyzed using an X-ray diffractometer (XRD, Rigaku DMAX 2400, Japan). Temperature and heat flow were calibrated using an Indium standard. Samples ranging from 5 to 10 mg were heated from 25 °C to 600 °C at a rate of 10 °C/min in a nitrogen atmosphere. Both thermogravimetric analysis (TGA) curves and differential scanning calorimeter (DSC) curves were obtained and compared for further analysis. The contact angle of PLCLxMS sheets was measured in sessile drop mode using a DSA100 (Krüss, Hamburg, Germany) following the method described by the manufacturer at 25 °C and 60% relative humidity. Four parallel samples were collected for each group.

### Mechanical properties

**Mechanical evaluation of MgSO$_4$ -reinforced PLCL samples.** During the radial compression test, PLCLxMS stents were positioned between two flat plates within a universal testing machine (Instron 5969, USA). The samples were compressed to 50% of their original diameter at a compression rate of 1 mm/min (Fig. 4A). Four parallel samples were tested for each group. The compressive strength of the stents was subsequently determined at this 50% compression point[115].

In the tensile mechanical property test, PLCLxMS 100 µm × 10 cm × 1 cm sheets were secured at both ends with clips in the same testing machine at room temperature. The sheets were stretched until they broke or reached 50% of their original length at a tensile rate of 1 cm/min. Four parallel samples were tested for each group. Subsequently, the Young's modulus was calculated using the following equation:

$$E = \frac{\sigma}{\varepsilon} = \frac{\mathbf{F}\nabla L}{AL_0} \tag{2}$$

Where:

$E$ is the Young's modulus (N/m$^2$),
$\sigma$ is the stress (N),
$\varepsilon$ is the strain (m/m),
$\mathbf{F}$ is the Load (N),
$\nabla L$ is the elongation of the sheet (m),
$L_0$ is the original length of the sheet (m),
A is the cross-sectional area of the sheet (m$^2$).

## Investigation of Mg²⁺ ions strengthening mechanisms

PLCL sheets were divided into two groups and immersed in deionized water and 1 mmol/L MgSO₄ solution, respectively, for 30 days at 37 °C. Four parallel samples were tested for each group. After air drying, the samples' hardness was determined using a digital Vickers microhardness tester (HMV-2T, Shimadzu Corporation, Japan) with a 0.98 N load and a 115-second dwell time. X-ray photoelectron spectroscopy (XPS) was performed using an XSAM600 instrument from the UK with Al Kα (1486.6 eV). The overview XPS spectrum was recorded from 0.08 to 1348.08 eV with an energy step of 1 eV, and binding energy was corrected using the C$1s$ (284.6 eV) as a reference. High-resolution O$1s$ spectra were acquired. Fourier-transform infrared (FTIR) spectra were also collected to elucidate the chemical groups of the samples.

## In vitro degradation study

An in vitro accelerated degradation study was conducted according to ISO 13781:2017 standard to evaluate the decomposition behavior of the composite stents in body fluid. The stents were immersed in 10 ml of phosphate-buffered solution (PBS) in 15 ml centrifuge tubes and incubated at 37 °C for 0, 7, 14, 28, and 56 days. Four samples were used for each time point. At each time point, the stents were removed from the tubes, the pH of the solution was measured, and the stents were then rinsed with distilled water and dried in a fume hood at room temperature for 24 h. After air drying, the samples were weighed and observed using SEM (S-4800, Hitachi, Japan) with an energy-disperse spectrometer (EDS) attachment after being sputtered with gold. The mechanical properties of the stent after immersion were assessed by the aforementioned radial compression test. Fourier-transform infrared (FTIR) spectra were also collected to elucidate the chemical groups of the samples.

## In vitro blood compatibility study

**Platelets adhesion test.** Fresh blood from New Zealand rabbits was centrifuged at 1000 x g for 15 min to obtain platelet-rich plasma (PRP) from the supernatant. PLCL, PLCL5MS, and PLCL10MS sheets, both before and after immersion in PBS for 7 days and drying (designated as PLCL-7D, PLCL5MS-7D, and PLCL10MS-7D) were placed in a 24-well plate. 0.4 mL of PRP was added to each well to ensure the samples were submerged. After 60 min of incubation at 39 °C, the PRP was removed, and the discs were gently rinsed three times with PBS. Platelets on the sheet surfaces were fixed using a 2.5% glutaraldehyde solution for 60 min, then dehydrated in gradient ethanol solutions (50%, 60%, 70%, 80%, 90%, 100%) for 10 min each. The sheets were then dried at 25 °C. The morphology of the platelets on the disc surfaces was observed using SEM. Three parallel samples were prepared for each material.

## Hemolysis rate analysis

Fresh blood was collected from healthy New Zealand rabbits, and 3.8 wt. % sodium citrate was added to the blood samples. The blood-to-citrate volume ratio was 9:1. The blood samples, mixed with sodium citrate, were diluted with PBS at a 4:5 blood-to-PBS ratio. The diluted blood samples were transferred into a centrifuge tube and immersed in 10 ml PBS at 39 °C for 30 min. Deionized water and PBS were used as the positive and negative control groups, respectively. The sheets were immersed in 10 mL PBS at 39 °C for 30 min, followed by the addition of 0.2 mL of blood. These mixtures were incubated in the alloy extracts at 39 °C for 60 min. The samples were then centrifuged, and the absorbance of the supernatant at 545 nm was measured using an enzyme-plate analyzer (Bio-RAD 680). Three parallel samples were prepared for each material.

## Simulation of PLCLxMS stent compress and substrate degradation

Mechanical properties and degradation behavior of PLCLxMS composite stents were simulated using COMSOL Multiphysics® software.

Initially, a meshed 3D model of the stent, based on the provided CAD data, was created to simulate mechanical properties. An external force was then applied to the stent's boundary to induce compression from the outside to the inside. Subsequently, the steady-state hysteresis equation of the stent was formulated using the plasticity model, as detailed below:

$$0 = \nabla S + \mathbf{F}_v \tag{3}$$

$$S = S_{intel} + S_{el}, \varepsilon_{el} = \varepsilon - \varepsilon_{intel} \tag{4}$$

$$S_{el} = C : \varepsilon_{el} \tag{5}$$

$$S_{intel} = S_0 + S_{ext} + S_q \tag{6}$$

$$\varepsilon = \frac{1}{2}[(\nabla u)^T + \nabla u] \tag{7}$$

$$C = C(E, v) \tag{8}$$

where:
$S$ is the strain (m/m),
$\mathbf{F}_v$ is the bulk force (N/m2),
$\varepsilon$ is the strain second-order tensor (m/m),
$u$ is the displacement field (m/m),
$E$ is the modulus of elasticity of the material (N/m²),
$v$ is the Poisson's ratio of the material.

The model of the dissolution chemical reaction process is divided into two segments. Initially, reaction engineering is utilized to simulate the intermittent reaction process in a mixed environment, assuming the absence of spatial dependence in the reaction. Subsequently, reaction engineering is coupled with the diffusion process of dilute substances in porous media to investigate the release of Mg²⁺ following the degradation of PLCL polymer chains. The transient mass balance equation for each substance is provided as follows:

$$\frac{\partial c_i}{\partial t} + \nabla \cdot (-D_{ik} \nabla c_i) = R_{ik} + R_{s,i} S_{sa} \tag{9}$$

Where:
$D_{ik}$ is the diffusion coefficient of substance $i$ in the corresponding medium $k$,
$c_i$ is the content of substance $i$ (m²/s),
$R_{ik}$ is the rate expression for a volumetric reaction involving only the substance $i$ in domain $k$ (mol/(m · s)),
$R_{s,i}$ is the surface reaction rate (mol/(m · s)),
$S_{sa}$ is the specific surface area of the porous stent (m²).

Magnesium ion transfer is governed by the diffusion equation. The sidewall boundary condition enforces zero mass flux, resembling an insulating state. At the stent's boundary, substance release is governed by reaction control. The stent-solvent interface maintains continuous material composition and flux, ensuring mass balance. The mass balance equation, expressed using Fick's law (without accounting for natural convection processes), is:

$$\frac{\partial c_i}{\partial t} = \nabla \cdot (D_i \nabla c_i) + R_x \tag{10}$$

Where:
$D_i$ is the diffusion coefficient of substance (m²/s),
$C_i$ is the concentration (mg/L),
$R_x$ is the reaction rate (mol/(m·s)),

The equation for the constant diffusion coefficient takes the following form:

$$\frac{\partial c_i}{\partial t} = D_i \frac{\partial C_i}{\partial x} + D_i \frac{\partial C_i}{\partial y} + D_i \frac{\partial C_i}{\partial z} + R_x \tag{11}$$

In the porous region, the reaction rate is expressed as:

$$R_{s,i} = R_i^s S_{area} \tag{12}$$

$$\frac{\partial c_{surf,i}}{\partial t} = R_i^s \tag{13}$$

Where:

$R_i^s$ is the reaction rate (mol/(m·s)),

$S_{area}$ is the surface area (m²),

$c_{surf,i}$ is the surface concentration where the substance diffusion coefficient is given according to previous experiment (mg/L),

$R_x$ is the rate of reaction generation (mol/(m·s)).

## Cell experiments

**In vitro cytotoxicity assays.** Human umbilical vein endothelial cells (HUVECs, purchased from Wuhan Prucell Life Sciences Co. (Wuhan, China)), human artery smooth muscle cells (HASMCs, purchased from Wuhan Prucell Life Sciences Co. (Wuhan, China)), and Hippocampal Tumor-22 (HT22, purchased from BeNaCultureCollection (Beijing, China)) were selected to assess the cytotoxicity of PLCLxMS composites according to the ISO-10993 standard[116]. Initially, extracts were prepared by immersing different PLCLxMS sheets in Dulbecco's Modified Eagle Medium (DMEM) supplemented with 10% fetal bovine serum and 1% penicillin/streptomycin. The magnesium concentrations in the extracts were measured using Inductively Coupled Plasma Optical Emission Spectroscopy (ICP-OES, iCAP6300, Thermo). The cells were then incubated at 37 °C with 5% $CO_2$ for 1, 3, and 5 days in either control medium (DMEM) or extract from different PLCLxMS sheets. Cell viability was evaluated using a standard CCK-8 assay, where cells were incubated with CCK-8 solution for 1 hour, and optical absorbance at 450 nm was measured using a spectrophotometer. Additionally, cells were stained using the LIVE/DEAD® assay after one day of culture, following the manufacturer's instructions, with living and dead cells stained by Calcein-AM and PI in green and red colors, respectively.

## In vitro neuroprotection assessments

To simulate ischemia/reperfusion injury in vitro, HT-22 cells underwent oxygen-glucose deprivation followed by reoxygenation (OGD/R) according to the ISO-10993 standard[117]. To induce hypoxia, cells were cultured in glucose-free medium for four hours at 37 °C in an incubator with 95% $N_2$/5% $CO_2$. Subsequently, the cells were reoxygenated for 24 h at 37 °C with 5% $CO_2$ in either regular culture medium or with extract from $MgSO_4$-loaded PLCL. Control groups were maintained in standard DMEM containing glucose under the same conditions. Viability, lactate dehydrogenase (LDH) release, reactive oxygen species (ROS) generation and $Ca^{2+}$ influx were then assessed.

Initially, cell viability was measured using the CCK-8 kit. Secondly, LDH release was quantified using an LDH assay kit; after incubating cell supernatants with the assay kit solution, the resulting color change was measured spectrophotometrically at 490 nm. Thirdly, intracellular ROS was detected using DCFH-DA: cells were cultured in serum-free medium supplemented with 10 μM DCFH-DA, then lysed with lysis buffer (50% methanol with 0.1 mg NaOH), and fluorescence at 488/525 nm was recorded using a fluorescence microplate reader. Next, intracellular $Ca^{2+}$ was measured using the Fluo-4 calcium assay kit; after treating cells with Fluo-4 AM solution, fluorescence intensity was measured using Image J software. Finally, neural cell apoptosis rate was detected by flow cytometry. Briefly, adherent cells were collected and washed twice. The cells were resuspended in 1× Binding Buffer and 100 μL of the suspension was added to the tube. After adding 5 μL of Annexin V-FITC and gently mixing, 10 μL of propyl iodide dye was added and suspension was mixed again to incubate at RT for 20 min. Then, use a flow cytometry machine to test within an hour later. As a supplementary experiment, Human Neuroblastoma Cell Line SH-SY5Y (SH-SY5Y, purchased from BeNaCultureCollection (Beijing, China)) cells were also treated under OGD/R and the cell viability was measured using the CCK-8 kit.

## Animal experiments

Male Sprague-Dawley (SD) rats (weighing between 280 and 320 g, aging between 8 and 12 weeks) were utilized in the investigation. The live experimental process was displayed in the Fig. 1A and the animal experiment setup for assessing brain integrity, cerebral blood flow (CBF), and behavioral testing was displayed in the Fig. S25.

## MCAO model and material implantation

Male SD rats were divided into five groups. The samples in this experiment were randomly assigned, and there was no considered control. The sample size is determined on 3Rs (Replace, Reduce and Refine) principle for animal experiments, and is kept to a minimum number while obtaining statistically significant data[118]. One group served as the healthy, sham-operated control, while others underwent the MCAO procedure with or without implantation of PLCL, PLCL5MS, and PLCL10MS wires (tMCAO+PLCL, tMCAO+PLCL5MS, and tMCAO+PLCL10MS groups). Three days prior to this procedure, rats received Aspirin (10 mg/kg/day) and Clopidogrel (7.5 mg/kg/day) until sacrifice. The MCAO procedure was performed as previously described[119]. Rats were anesthetized with 1-3% isoflurane, followed by right MCA occlusion using a filament. The filament was withdrawn after 2 h to allow reperfusion of the ischemic hemisphere and get stable injury model[120]. The filament was withdrawn after 2 h to allow reperfusion of the ischemic hemisphere. For the pMCAO group, the filament was not withdrawn. Then, 1 cm long PLCLxMS wires were inserted through an opening in the ECA into the CCA.

An additional group (intravenous injection, denoted as the IV group) was established by immersing the PLCL10MS stents in 10 ml of saline. The extracts were collected and replaced with fresh saline on days 1, 3, 5, and 7. At each time point, the extracts were administered to tMCAO rats via the tail vein, after being disinfected and filtered. These rats were designated as the tMCAO+$Mg^{2+}$ from the PLCL10MS IV injection group. Some rats underwent 14 days of continuous behavioral testing, while others were sacrificed on day seven for brain removal and Nissl staining.

## Neurological behavioral tests

To assess the neurological deficits in the rats, four different neurological behavioral tests were conducted. The open field test was performed 7 days post-surgery, and the Zea-Longa scoring, rotarod, and adhesive touch removal tests were conducted before surgery and on days 1, 3, 5, 7, 10, and 14 post-surgeries.

Zea-Longa scores ranged from 0 (no brain damage) to 4 (unconscious and unable to walk independently). In the rotarod test, rats were placed on an accelerating rotating rod, with the speed gradually increasing from 4 to 40 rpm over five minutes. For the adhesive touch removal test, a small piece of tape was applied to the left paw of each rat, and the average time taken to contact and remove it was recorded. If the tape was not removed within two minutes, the time was recorded as two minutes. For the open field test, each rat was placed in the center of an open field for five minutes in a quiet room. Animal behavior was recorded using a computer-connected video monitoring device, and the video was analyzed to measure walking distance and time spent in different areas.

## Infarct volume evaluation

After transcardial perfusion with PBS, rat brains were removed and sliced into 2 mm thick sections. The slices were then stained with triphenyl tetrazolium chloride (TTC, 2% wt./vol. in PBS) for 20 minutes at 37 °C. Photographs of each slice were taken, and infarct volume ratios were calculated using Image J. This outcome was expressed as the ratio of (contralateral hemisphere's area-ipsilateral hemisphere's non-infarcted area) to the contralateral hemisphere's area.

## Nissl staining

After extraction, brains were fixed in 10% formaldehyde for a full day. Specimens were then embedded in paraffin and sectioned, followed by staining with Nissl staining solution. The stained sections were photographed, and the number of Nissl-positive cells was counted using Image J. The Nissl body ratio was calculated by dividing the number of Nissl-positive cells in the ipsilateral hemisphere by the number of Nissl-positive cells in the contralateral hemisphere.

## Brain edema

Initially, the brain wet weights were measured, followed by drying the rat brain tissue for 24 h at 90 °C in a constant temperature oven, and then measuring the brain dry weight. The proportion of water in the brain tissue is calculated as (1-brain dry weight/brain wet weight) ×100%.

## Evans blue leakage

A 2% Evans blue dye (EB, 2% wt. /vol in PBS, 3 mL/kg) solution was injected into the tail vein. Two h later, after cardiac saline perfusion, the brain was removed to demonstrate EB extravasation. Additionally, the ischemic half of the brain tissue was homogenized into a cell suspension with the addition of 1 mL formamide solution. The sample was then incubated in a water bath at 60 °C for 24 h. After centrifugation at 5000 rpm for 10 minutes, the optical density of the supernatant at 635 mm was measured using a spectrophotometer, and the EB content was calculated according to the standard curve.

## Measurement of cerebral blood flow

Relative cerebral blood flow (CBF) was determined using laser speckle imaging (LSI, RWD Science Co., China). The rats, anesthetized with 1%-3% enflurane after initial 5% enflurane administration, had their cranial skin shaved and disinfected with iodophor. The animal was then positioned in a prone posture with its head secured in a stereotaxic device. A longitudinal incision was made in the cranial skin. The skull was carefully thinned from bregma to lambda under a dissecting microscope using a dental drill until the LSI could clearly detect CBF. CBF was measured for at least five minutes in each hemisphere of the brain during the experiment. Color bands representing perfusion units were used to visualize CBF, and the average CBF perfusion units for rats in both hemispheres were calculated. The CBF on the damaged side was expressed as a percentage of the value on the contralateral side.

## In vivo PLCL degradation experiment

PLCL, PLCL5MS, and PLCL10MS wires were retrieved 7 days after implantation, with three wires collected at each time point. The wires were pressure-perfused with saline to remove blood and then immersed in 2.5 % glutaraldehyde for 12 h. Subsequently, dehydration was performed using graded ethanol (20%, 40%, 60%, 80%, 95%, 100%) with 30 min for each concentration. Surface morphology of the wires was examined using a scanning electron microscope (SEM).

## Ex vivo $Mg^{2+}$ distribution

The approach employed for identifying trace metals released in vivo was derived from the work of Matusiewicz[121,122]. Blood samples were taken from the right common carotid artery (CCA) on days 1 and 7 following implantation, utilizing a polymer needle. Subsequent to euthanasia, organs such as the infarcted contralateral brain (ICB) and infarcted ipsilateral brain (IIB) were excised with Teflon-coated tweezers and ceramic scissors, then weighed and transferred into Teflon digestion vials. The samples were then digested using HNO3 and H2O2 in a microwave system, and the magnesium concentration was analyzed through ICP-OES (iCAP6300, Thermo).

## Ex vivo ROS detection

The brain tissue was blended in a 1:10 w/v solution of RIPA buffer. Following the homogenization process, the samples of brain tissue were centrifuged at 12,000 rpm for a duration of 20 min at a temperature of 4 °C. According to Genmed Scientifics Inc., USA, Lucigenin was mixed with the resulting supernatant. After allowing a 15-minute acclimatization period, luminescence readings were taken every second for a total of 10 seconds using a luminometer (ThermoFisher Scientific, USA), with results expressed in relative light units per second.

## Ex vivo $Ca^{2+}$ detection

In accordance with the instructions provided by the manufacturer (Beyotime, China), the concentration of $Ca^{2+}$ in brain tissue was assessed. First, small fragments of the brain tissue were prepared, and a lysis solution was introduced at a dosage of 1000 μL for every 20 mg of tissue. This mixture underwent homogenization, followed by centrifugation at 10,000 to 14,000 × $g$ for 5 min at a temperature of 4 °C to retrieve the supernatant. Subsequently, the supernatant was allowed to incubate with the working solution for $Ca^{2+}$ testing for a duration of 10 min at ambient temperature. The absorbance was recorded at 575 nm using an enzyme-linked immunosorbent assay method, and the concentration of $Ca^{2+}$ was calculated using a standard curve for calibration.

## PLCL10MS stent implantation in rabbit CCA model and analysis method

Forty male New Zealand rabbits (weighing between 3 and 3.5 kg, aging between 4 and 6 months) were utilized in the investigation; ten served as the healthy, sham-operated control, fifteen were successfully implanted with PLCL stents and fifteen with PLCL10MS stents. The stents were sterilized by ethylene oxide. All experimental procedures received approval from the Capital Medical University's Institutional Animal Investigation Committee and were conducted in accordance with the National Institutes of Health's standards for the Care and Use of Laboratory Animals. During implantation, a PLCL or PLCL10MS stent, was deployed in the common carotid artery (CCA) of New Zealand rabbits. To do this, after 12 h of fasting, anesthesia was performed by intravenous injection of 1 ml/kg pentobarbital. Then the rabbit was fixed in a supine position and the stent was delivered to CCA via minimally invasive surgery from the femoral artery. The stents were deployed with the recommended pressure of 12 - 16 atm for PLCL and PLCL10MS stents. Digital Subtraction Angiography (DSA) were conducted immediately after the balloon deflation to confirm the patency of the blood flow and the correct placement of the stent. The arteries contained the stents at each time point (7 days and 1 month) were harvested for subsequent analysis. And some samples were embedded in paraffin and sliced into 5 μm sections, and then stained with hematoxylin-eosin (HE). Others were fixed with 2.5% glutaraldehyde and left for 12 h. Subsequently these were dehydrated using a series of gradient ethanol solutions of 30%, 50%, 70%, 80%, 90%, 100%, and 100% with each step lasting 15 min. The targeted artery segments were then longitudinally bisected and coated with a thin layer of gold before being examined using a scanning electron microscope (SEM). Blood from rabbit CCA near the stent implantation position was collected on days 1 and 7 post-implantation and Mg concentration was determined via ICP-OES (iCAP6300, Thermo) after being digested with HNO₃.

## Statistical analysis

All results were expressed as mean ± standard deviation. SPSS 17.0 was used for statistical analysis. Differences between two groups were compared using the two-tailed unpaired Student's t-test, and differences in means among multiple groups were analyzed using one-way ANOVA followed by the Tukey/Games-Howell post hoc correction. Data pertaining to various time points and groups were subjected to the two-ways ANOVA analysis, which was also followed by the Bonferroni post hoc correction.

## Reporting summary

Further information on research design is available in the Nature Portfolio Reporting Summary linked to this article.

## Data availability

All relevant data that support the findings of this study are available within the article and Supplementary Information. All data are available from the corresponding authors upon request. Source data are provided with this paper.

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

## Acknowledgements

This work was supported by CHF-brain (2024-3-2062, ML), National Key R&D Program of China (Grant No. 2021YFC2400700, YFZ), Excellent Youth Fund of Capital Medical University (B2305, ML), the Non-profit Central Research Institute Fund of Chinese Academy of Medical (2023-JKCS-09, ML; 2023-JKCS-13, MJ), Beijing Municipal Administration of Hospitals Clinical Medicine Development of Special Funding Support from Yangfan Project (YGLX202325, ML), Research Funding on Translational Medicine from Beijing Municipal Science and Technology Commission (Z221100007422023, ML), Beijing Association for Science and Technology Youth Talent Support Program (BYESS2022081, ML), National Natural Science Foundation of China (82102220, ML; 82027802, XJ), Collaborative innovation project by Chinese Institutes for Medical Research (CX25XT02, MJ), Grants from the National Natural Science Foundation of China (82402444, MJ) and Beijing Natural Science Foundation (7244510, MJ).

## Author contributions

H.Z., Y.Z. and L.S. contributed equally to the work. Conceptualization: H.Z., Y.Z., M.L., and Y.F.Z. Methodology: H.Z., Y.Z., L.S., and Y.S. Investigation: H.Z., X.C., C.W., G.S., and M.L. Visualization: H.Z., Y.Z., B.S., and Z.X. Funding acquisition: X.C., H.S., Q.L., X.J., and M.J. Project administration: C.W., X.J., M.J., M.L., and Y.F.Z. Writing – original draft: H.Z., Y.Z., and M.L. Writing – review & editing: X.J., M.J., M.L., and Y.F.Z.

## Competing interests

The authors declare no competing interests.
