## [Transparent Peer Review file · Nature Communications]

Mechanically robust neuroprotective stent by sequential Mg ions release for ischemic stroke therapy

Corresponding Author: Professor Yufeng Zheng

Version 0:

Reviewer comments:

Reviewer #1

(Remarks to the Author)

Key results:

In this study, Hongkang Zhang et al., the authors have developed a biodegradable Mg²⁺ releasing stent, to be used to confer post-stroke neuroprotection. The authors have very extensively characterised the composition and surface morphology of their PCLC (Poly(lactide-co-ε-caprolactone) based stent and have studied the dynamics of Mg²⁺ release and polymer degradation in vitro. The authors then used their Mg²⁺ releasing biodegradable material in vitro to test its cytotoxicity and neuroprotection property. The authors then used their Mg²⁺ releasing biodegradable material to construct wires, which were tested for the improvement of infarction area, blood-brain barrier integrity and cerebral blood flow in in vivo middle cerebral artery occlusion (MCAO) rat model and showed that they confer long-term post-stroke neuroprotection, including improved behavioural test in MCAO rat model. Finally, the authors made a spiral stent from their Mg²⁺ releasing biodegradable material and showed that it reduced infarct volume on the first day after implantation.

Major concern:

The authors claim to have produced first biodegradable neuroprotective brain stent. Given that long-term neuroprotection (both in vitro and in vivo) by Mg²⁺ ions, and functional biodegradable Mg-alloy stents, have been earlier reported extensively, these findings significantly undermine the novelty of their study. Furthermore, the fact that the authors have shown long term in vivo neuroprotection with wires fabricated from their material (and not stent), also renders the claim of being the first biodegradable neuroprotective stent somewhat exaggerated.

With their spiral stent, the authors have only shown a reduction in infarct volume. It is not clear from the text, and the figures, if the authors have actually tested only wires (not shown in the manuscript) or the cylindrical stent shown in Figure 7A for their long-term studies with behavioural tests? A solid wire (not shown) vs unopened stent (figure 7A, left) vs a spiral stent (S11A) have very different dynamics in terms of their placement, stability at the insertion site, strength, intactness and biodegradation. Therefore, for their claim regarding first biodegradable neuroprotective brain stent, the authors must use their open stent to show long-term post-stroke neuroprotection, show the long-term structural integrity and the in vivo Mg²⁺ release of their used stent.

Minor concerns:

1) S9 D: please specify the respective stainings on the figure?

2) Is there a specific reason for using human lines for endothelial cells and smooth muscle cells but mouse cell line for neurons (instead of say SH-SY5Y human cell line for neurons)?

3) Can the authors comment on why there is no additional burst of Mg²⁺ once the PLCL is hydrolysed to release the MgSO₄ particles from the inside of the stent?

4) The paper is underlining the role of Mg²⁺ ions in neuroprotection. The authors have also demonstrated that their material secretes only Mg²⁺ ions. However, in their previous study, Mg²⁺ alone has been shown to be insufficient to reduce infarct size (reference 6). It is not clear how the released Mg²⁺ from their fabricated wires/stent (at different concentrations) is able to produce contradicting results.

Reviewer #2

(Remarks to the Author)

Current treatment options for ischemic stroke are limited to thrombolysis and thrombectomy, targeting reperfusion of ischemic tissue. However, ischemia-reperfusion injury per se may augment tissue injury and remains a fundamental problem that needs to be addressed. In their study Zhang and colleagues aim to introduce a Mg²⁺ eluting stent by incorporating neuroprotective MgSO₄ particles into a PLCL (substrate using 3D printing technology). They show that the release of Mg²⁺ provides sequential neuroprotection that aligns with the treatment window for acute ischemic stroke. However, the advantage of release Mg²⁺ by the stent compared to the class IP or IV injection is not clear. Moreover, Magnesium neuroprotection in acute stroke in clinical set is still controversial, some clinical studies show no effect (i.e. DOI: 10.1056/NEJMoa1408827; DOI: 10.1161/STROKEAHA.118.021916), the authors do not refer to any of these studies. More comments include:

1. This approach would have more impact if the PLCL10MS had bigger neuroprotective effect than Magnesium injected IP or IV as in previous studies, one more group (MCAO+stand+Mg²⁺ injected IP/IV) would have clarified the advantage of this technology.
2. TTC staining is not ideal to quantify ischemia 7 days post-stroke, Nissle or H&E stainings are more appropriate.
3. "In vivo" ROS detection is not really an in vivo quantification but ex-vivo. A different approach should be used to really measure the ROS levels in vivo. Same for "in vivo" Mg²⁺ distribution..
4. Most of the behavioral seem to have a significant difference only at 14days in between groups, however the authors hypothesized that stands might have acute and long term functional recovery. Why are the PLCL5MS and PLCL10MS only beneficial after 2 weeks if ischemic volume, BBB leakage etc are already neuroprotective at 1 week?
5. The authors also hypothesized that stents may offer sequential neuroprotection to remote ischemic brain tissues, however no mechanisms are shown, no trigger is hypothesized for remote neuroprotection.
6. Number of animals per group in Fig 7 is only showed in panel c.
7. How was the sample size measured?
8. Some statistics should use two-ways Anova.
9. In Methods section, laser speckle imaging acquisition and analysis is not included. Moreover, because carbon dioxide (CO₂) state might affect the results, a systemic arterial blood-gas measurement needs to be performed at the end of each experiment to record pCO₂ and blood pressure (magnesium could reduce blood pressure).
10. To mimic a post-thrombotic event with reperfusion damage, a clot-model would be more appropriate than the filament model. Comparing the effect of PLCL10MS in a permanent model with no reperfusion would address the effect on the reperfusion injury.

Reviewer #3

(Remarks to the Author)

This is a study of a novel Mg-eluting PLCL stent for ischemia-reperfusion neuroprotection with an evaluation of its mechanical, surface, and structural properties, elution characteristics, and in vitro and in vivo biologic effects. The authors characterized the novel biodegradable stent and demonstrated its potential utility in vitro and in vivo. There are a few queries that require consideration:

There are minor grammatical errors sprinkled throughout the manuscript including "This stage is characterized the penetrating of water into the PLCL matrix and the outward diffusion of Mg²⁺ from within the stent", and "And the PLCLxMS sheets exhibited a significant increase...". Also, CAD is used both as an abbreviation for computer-aided design as well as coronary artery disease.

In the X-ray diffractometer data in Figure 2D, can the authors speculate why there is a MgSO₄ (101) peak for the control (PLCL group)? How was the potential for Mg contamination on PLCL alone stents controlled if using the same 3D extrusion deposition printer?

As per Figure S6, surface roughness increases due to magnesium dissolution within 7 days. Can the authors speculate what this early and progressive roughness will do to the thrombogenic potential of the stent over time?

It is interesting that in the in vitro experiments, Mg continued to be released up to 56 days. However, in the in vivo studies, Mg concentration reduced to normal levels by day 7. Is the reduction of Mg conc in the IIB by day 7 related to the completion of Mg dissolution, or could endothelialization of the stent surface play a role here? In other words, could endothelialization (which can take place over a similar timeframe in rats) affect the dissolution of Mg from the stent and subsequent effectiveness since it may prevent the Mg from being released and reaching downstream targets? Did the authors histologically evaluate the spiral stent explants for endothelialization after 7 days.

The authors provide a reference for the neuroprotective effect of Mg being dose dependent, although there was no dose dependent effect on in vitro neuroprotection shown in vitro beyond 10% MgSO₄. How can the authors reconcile this?

The authors write "As shown in Fig. 3D and Fig S6, the MgSO₄ particles embedded in the stent surface were completely dissolved after 7 days of immersion, leaving behind cavities at their original locations. However, MgSO₄ particles remained inside the stent struts obtained from the cross-sectional morphology." Since Mg levels in the IIB normalized by day 7, one could argue that this was all surface Mg and the inside Mg was not required to produce an effect in vivo. What is the importance, then, of the internal Mg as opposed to simply a stent surface coating of Mg?

Although numerous studies were completed on PLCLxMS stent degradation, including its effect on the mechanical properties of the stent, this was performed in the context of Mg release and did not address PLCL stent degradation over the longer term. The full time course of PLCL degradation was not addressed but should be mentioned or referenced.

N values for number of animals and in vitro studies should be included.

Version 1:

Reviewer comments:

Reviewer #2

(Remarks to the Author)

The authors have addressed my questions

Reviewer #3

(Remarks to the Author)

I commend the authors for their additional studies, responses, and the edits made to address comments made in the first manuscript review. Specifically, all of my comments/concerns have been adequately addressed in the revised version of this manuscript.

Reviewer #4

(Remarks to the Author)

This paper explores an important topic in the treatment of acute ischemic stroke, particularly the therapeutic challenges encountered by patients with large - vessel occlusion after thrombectomy. In the work, the prepared PLCL scaffolds with different concentrations of MgSO₄ using 3D - printing technology, aiming to explore their potential application in alleviating ischemia - reperfusion injury.

Overall, this study provides one potential way for the later - stage treatment of stroke by evaluating the performance of the scaffold, the release mechanism of magnesium ions, and the neuroprotective effects.

However, the some key points still require the further resolution:

1. Please clearly explain the related clinical pathological mechanism of acute ischemia stroke to support the rationality of the staged-release of Mg²⁺ in time and dose, and further state the clinical application significance that comes with the staged-release.

2. Lines 101 - 140: please introduce the advantages or the innovation of PLCL. The used When introducing the properties of experimental materials, some comparative analysis with other commonly used materials can be added to highlight the advantages and characteristics of the materials in this study. For example, when introducing the properties of PLCL, a comparison with other biological scaffold materials can be made.

3. Lines 142 - 169: please clearly state or display the innovative characteristics in experimental design. For some complex results, such as the three stages of Mg²⁺ release, charts or diagrams can be used for more intuitive display, and the key characteristics of each stage can be emphasized in the text description.

4. Line 335: The rationality of the experimental sample size should be explained, including the basis for choosing the current sample size and whether the sample size is sufficient to support the research conclusions.

5.Lines 648 - 655: please state the safety analysis of the materials, including the research on long - term safety and potential adverse reactions.

6. Line 858: the indicate removal after 2 hours, but there is no basis or timeline for fiber removal, and the corresponding clinical window's pathological mechanism is not provided.

7. Experimental scope: Although the study shows that the scaffold can effectively release magnesium ions at different stages, the long - term effects and the applicability to different types of stroke have not been sufficiently explored. In the future, the experimental scope can be expanded to further verify the long - term efficacy and safety of the scaffold. Further research on the degradation products of the scaffold after long - term implantation in the body and their impact on surrounding tissues, including possible inflammatory reactions and tissue repair processes, is needed.

8. Scaffold degradation characteristics: Although the degradation performance of the scaffold was mentioned, the long - term impact of the degradation rate on the compatibility with brain tissue was not discussed in depth. Especially whether the release mechanism of magnesium ions and the degradation process of the scaffold will have a negative impact on brain tissue is worth further research.

9. Production cost and application feasibility of the scaffold: The cost issue of 3D - printed scaffold technology and the feasibility of its actual clinical application, especially the promotion for large - scale clinical application, were not discussed in detail in the article. As a new type of therapeutic method, how to reduce the cost and ensure its wide clinical application is an important issue that needs to be considered.

Version 2:

Reviewer comments:

Reviewer #4

(Remarks to the Author)

I have no more comments.

Response to Reviewer 1

Reviewer #1 (Remarks to the Author):

Key results:

In this study, Hongkang Zhang et al., the authors have developed a biodegradable Mg^{2+} releasing stent, to be used to confer post-stroke neuroprotection. The authors have very extensively characterised the composition and surface morphology of their PCLC (Poly(lactide-co- ϵ -caprolactone) based stent and have studied the dynamics of Mg^{2+} release and polymer degradation in vitro. The authors then used their Mg^{2+} releasing biodegradable material in vitro to test its cytotoxicity and neuroprotection property. The authors then used their Mg^{2+} releasing biodegradable material to construct wires, which were tested for the improvement of infarction area, blood-brain barrier integrity and cerebral blood flow in in vivo middle cerebral artery occlusion (MCAO) rat model and showed that they confer long-term post-stroke neuroprotection, including improved behavioural test in MCAO rat model. Finally, the authors made a spiral stent from their Mg^{2+} releasing biodegradable material and showed that it reduced infarct volume on the first day after implantation.

Major concern:

1. The authors claim to have produced first biodegradable neuroprotective brain stent. **【Q1】** Given that long-term neuroprotection (both in vitro and in vivo) by Mg^{2+} ions, and functional biodegradable Mg-alloy stents, have been earlier reported extensively, these findings significantly undermine the novelty of their study. **【Q2】** Furthermore, the fact that the authors have shown long term in vivo neuroprotection with wires fabricated from their material (and not stent), also renders the claim of being the first biodegradable neuroprotective stent somewhat exaggerated.

[Reply to Q1]: Dear reviewer, thank you for your comment. We do agree with you that the researches on the neuroprotection of Mg^{2+} and the biodegradable Mg-alloy stents have been earlier reported extensively.

As for the Mg^{2+} neuroprotection, most of the reported studies focus on the effects of administration route (intraperitoneal, intravenous, arterial or intracranial supplementation), administration time and magnesium dose on different stroke animal models (focal cerebral or global cerebral ischemia). However, the studies of continuously release of Mg^{2+} into ischemic brain tissues are barely reported.

As for the biodegradable Mg-alloy stents, most biomaterial scientists and manufacturers have only focused on the cardiovascular tissue and blood compatibility of stent materials, as well as the mechanical optimization of stent structures for heart vessels. However, the studies of biodegradable Mg-containing cerebral stents are still in early stages.

Our work is a proof-of-concept study on “Mg ions eluting biodegradable neuro-protective stents”. This concept is premised on the hypothesis that locally released Mg^{2+} from $MgSO_4$ particles reinforced 3D print PLCL stent could offer neuroprotection against reperfusion injury in distant cerebral ischemic tissues.

The novelty of our study mainly lies in the following three points.

(1) The staged-release of Mg^{2+} is found to may provide sequential neuroprotection effect that aligns with the treatment window for acute ischemic stroke. The release of Mg^{2+} exhibits three distinct stages: Burst (<1 day), Fast (1~7 day), and Stable (7~56 day).

(2) A novel MgSO₄-particle/Mg²⁺-ions combined-mechanical reinforcement mechanism was introduced. Because of the MgSO₄ particle strengthening effect, the radial strength of the composite stent was increased compared with that of pure PLCL stent. Interestingly, a Mg²⁺ strengthening mechanisms during stent degradation was observed and verified in our study.

(3) A pioneering *in vivo* bioactivity evaluation method was proposed by using a transient middle cerebral artery occlusion (MCAO) rat model to emulate human brain ischemia/reperfusion injury. This method is applicable to the *in vivo* neuro-compatible study of all degradable materials that being potentially used as brain implant.

Therefore, We added the description of “Previous research on Mg²⁺ neuroprotection primarily focuses on the effects of administration route (intraperitoneal, intravenous, arterial, or intracranial supplementation), administration timing, and Mg²⁺ dose in different stroke models (focal cerebral or global cerebral ischemia) [72, 80]. The objectives of studies on biodegradable Mg-alloy stents are to improve mechanical performance, endothelialization rate, and control substrate corrosion rate through alloying and surface coating (summarized in Table S2) [81-91]. However, in our proof-of-concept study, a novel Mg²⁺-eluting biodegradable neuroprotective stent was proposed. The staged release of Mg²⁺ was designed to provide sequential neuroprotective effects aligned with the treatment window for AIS, verified using a tMCAO rat model to replicate human brain ischemia/reperfusion injury. Moreover, the radial strength of the composite stent was increased compared to the pure PLCL stent, and a novel MgSO₄-particle/Mg-ions combined-mechanical reinforcement mechanism was introduced to explain this increase. This study may inform the design of novel biodegradable stents for brain disease treatment.” in the added section of Comparison with previous studies in Discussion of the revised manuscript.

Table S2 Summary of typical studies of biodegradable Mg alloy stents

Material	Implant type	Animal model	Function	Functional Method	Reference
AZ31B	Mg-alloy stents	Porcine coronary arteries	Improving the corrosion resistance	Chemical conversion coating and biodegradable PDLLA coating	2020 [81]
AZ31	Mg-alloy stents	NA	Improving the corrosion resistance	PBAT coating	2022 [82]
AZ31	Mg-alloy stents	SD rats abdominal aorta	Anticorrosive and antithrombotic function	PDDA incorporated multilayer coating	2021 [83]
AZ31B	Mg-alloy stents	Rabbit Infrarenal abdominal aorta	Preventing restenosis	P(LA-TMC) coating carrying sirolimus	2011 [84]

Mg-2.0Zn-0.5Y-0.5Nd	Mg-alloy stents	Porcine coronary artery	Improving corrosion resistance and othelialization	APTES treated silane coating and rapamycin-eluting PLGA layer	2016 [85]
Mg-2.5Nd-0.21Zn-0.44Zr	Mg-alloy stents	Rabbit's abdominal artery	Improving corrosion resistance	Design of the alloy composition	2017 [86]
Mg-2.5Nd-0.21Zn-0.44Zr	Mg-alloy stents	Porcine coronary artery	Improving corrosion resistance and inhibiting neointima proliferation	Rapamycin-eluting PLLA coating and protective MgF ₂ layer	2017 [87]
WE43	Mg-alloy stents	Porcine coronary artery	Improving machanical performance	Design of the alloy composition	2008 [88]
DREAMS (WE43)	Mg-alloy stents	Human Coronary artery	Improving corrosion resistance and inhibiting neointima proliferation	PLGA coating with paclitaxel	2013 [89]
DREAMS 2G (WE43)	Mg-alloy stents	Human Coronary artery	Improving corrosion resistance and inhibiting neointima proliferation	PLLA coating with sirolimus	2020 [90]
Mg-8.5Li	Mg-alloy stents	Porcine iliac artery	Improving mechanical performance	Design of the alloy composition	2021 [91]

[Reply to Q2]: Dear reviewer, we do agree with you that our experimental design is not perfect enough, and we demonstrated the *in vivo* neuroprotection of the materials in the form of wires but not stent. There are two reasons for this study design.

(1) Refer to the previous experience in stent design, the biological evaluation of new stent materials mainly starts with wire and then proceeds to stents.

(2) Our study focuses on the neuroprotection of Mg²⁺ ions on ischemic stroke with reperfusion injury. The rodent MCAO (middle cerebral artery occlusion) models is a widely used stroke models, but not suitable for stent implantation. As for large stroke animal models (swine and sheep models, including dogs and rabbits) demonstrate a significant collateral circulation facilitated by maxilla-carotid artery and meningocerebral anastomoses, as well as a large draining cerebral vein capable of reverse flow. This anatomical complexity can reduce the reproducibility of infarcts. Moreover, testing of functional outcome is not as widely applied in large animal models as in rodent models, in part, due to study constraints (such as the surgical approach) and ethical considerations.

Therefore, we have added the description of “One limitation of our research is that the *in vivo* neuroprotection evaluation of PLCLxMS was conducted using wires, rather than stents. In reference to the design of biodegradable metal and polymer stents, the safety analysis of these materials typically begins with wires [92, 93, 94] and progresses to stents[88, 95, 96]. For *in vivo* neurological analysis, the rodent MCAO model is widely used for ischemic stroke, but it is not suitable for stent implantation [97]. The anatomical complexity of large stroke animal models (swine, sheep, dog, and rabbit) reduces the reproducibility of infarcts, and neurological function assessment methods are not as widely applied in these models as in rodent models [98].” in the added section of Comparison

with previous studies in Discussion of the revised manuscript.

2. With their spiral stent, the authors have only shown a reduction in infarct volume. **【Q1】** It is not clear from the text, and the figures, if the authors have actually tested only wires (not shown in the manuscript) or the cylindrical stent shown in Figure 7A for their long-term studies with behavioural tests? **【Q2】** A solid wire (not shown) vs unopened stent (figure 7A, left) vs a spiral stent (S11A) have very different dynamics in terms of their placement, stability at the insertion site, strength, intactness and biodegradation. Therefore, for their claim regarding first biodegradable neuroprotective brain stent, the authors must use their open stent to show long-term post-stroke neuroprotection, show the long-term structural integrity and the *in vivo* Mg²⁺ release of their used stent.

[Reply to Q1]: Dear reviewer, thank you for your comment. We apologize for the misunderstanding caused by the unclear description in the text. We actually only used PLCLxMS wires for their *in vivo* neuroprotection analysis in MCAO rat models, except for the results in Fig. S11.

Therefore, we have modified the description as “ PLCL5MS and PLCL10MS wires were selected as stent implant alternatives for further *in vivo* experiments using MCAO rat models, with pure PLCL wires serving as the control group (Fig. 7A). **All the *in vivo* neurological analysis were conducted by implanting PLCLxMS wires into the tMCAO models.** ” and “ The PLCL10MS spiral stents still provided neuroprotection against I/R injury on the first day after implantation (Fig. S14C), **further demonstrating the *in vivo* neuroprotective effects of PLCLxMS, not only in the form of wires but also spiral stents.** ” in the section of *Ex vivo* neuroprotection analysis in the revised manuscript.

[Reply to Q2]: Dear reviewer, we do agree with you that different material configurations have very different dynamics in terms of their placement, stability at the insertion site, strength, intactness and biodegradation.

The reason of using wires for rat studies mainly lies in the following two points.

(1) As mentioned above, due to the reproducibility of infarct size, procedural complications, and functional outcome assessment, rodent ischemic stroke models are well recognized and widely used. The NHP models are most similar to human patients but are not frequently used due to the problems of ethics, logistics, and costs.

(2) The catheter-based intervention could not be performed in a rodent model. We found that it was hard to print a small stent to meet the size of the common carotid artery in rats. Therefore, we chose wires as a substitute for the stent and systematically studied the neurological function evaluation of rats.

The main purpose of curling the wires to form spiral stents, as shown in Fig.S11, is to further demonstrate the *in vivo* neuroprotective effects of PLCLxMS, not only in the form of wires but also spiral stents.

According to your suggestions, we conducted two supplementary experiments.

Firstly, based on your comments and the feasibility of animal study, we had tried to make a rabbit stroke model by using suture occlusion method, but we did not make it (as shown in the Figure R1). Further literature review suggests that the cerebral blood supply is more significantly

compensated by contralateral and posterior circulation in rabbits than that in rats, and simply occluding the opening of the middle cerebral artery cannot achieve a cerebral ischemia effect similar to that of rats [N.V. Phan, E.M. Rathbun, Y. Ouyang, S.T. Carmichael, T. Segura, *Biology-driven material design for ischaemic stroke repair*, *Nature Reviews Bioengineering* 2(1) (2024) 44-63].

Figure R1. TTC staining of rabbit brain slices for cerebral ischemia modeling attempt

Secondly, according to your kindly suggestion, we implanted the stent into the CCA of healthy rabbit models to study the structural integrity and the *in vivo* Mg²⁺ release of the opened stent.

Therefore, we have added the method description of “Forty male New Zealand rabbits weighing between 3 and 3.5 kg were utilized in the investigation; ten served as the healthy, sham-operated control, fifteen were successfully implanted with PLCL stents and fifteen with PLCL10MS stents. The stents were sterilized by ethylene oxide. All experimental procedures received approval from the Capital Medical University's Institutional Animal Investigation Committee and were conducted in accordance with the National Institutes of Health's standards for the Care and Use of Laboratory Animals. During implantation, a PLCL or PLCL10MS stent, was deployed in the common carotid artery (CCA) of New Zealand rabbits. To do this, after 12 h of fasting, anesthesia was performed by intravenous injection of 1 ml/kg pentobarbital. Then the rabbit was fixed in a supine position and the stent was delivered to CCA via minimally invasive surgery from femoral artery. The stents were deployed with the recommended pressure of 12~16 atm for PLCL and PLCL10MS stents. Digital Subtraction Angiography (DSA) were conducted immediately after the balloon deflation to confirm the patency of the blood flow and the correct placement of the stent. The arteries contained the stents at each time point (7 days and 1 month) were harvested for subsequent analysis. And some samples were embedded in paraffin and sliced into 5 μm sections, and then stained with hematoxylin-eosin (HE). Others were fixed with 2.5% glutaraldehyde and left for 12 h. Subsequently these were dehydrated using a series of gradient ethanol solutions of 30%, 50%, 70%, 80%, 90%, 100%, and 100% with each step lasting 15 min. The targeted artery segments were then longitudinally bisected coated with a thin layer of gold before examined using a scanning electron microscope (SEM). Blood from rabbit CCA near the stent implantation position was collected on days 1 and 7 post-implantation and Mg concentration was determined via ICP-OES (iCAP6300, Thermo) after digested with HNO₃,” in the section of PLCL10MS stent implantation in rabbit CCA model and analysis in the revised manuscript.

We added the results of “The optical morphology of the PLCL and PLCL10MS stents during crimp and expansion are shown in Fig. S19. Digital Subtraction Angiography images (DSA) in Fig. S20 show that both of the PLCL and the PLCL10MS stents were successfully implanted and no stenosis were observed at 7 days and 1 month. Combined with the SEM (Fig. S21) and HE staining (Fig. S22) results, the PLCL and PLCL10MS stents struts in rabbits CCA were partly covered by the neointima at 7 days and were fully embedded by intima at 1 month. The concentration of magnesium at 1 day in CCA blood of PLCL10MS stent group was higher than that of the sham and PLCL stent group, but with no statistical difference at 7 day (Fig. S23).” in the section of

Implantation result and histological analysis of PLCL10MS stent in rabbit CCA model in the revised manuscript.

Fig. S19.

Optical images of the PLCL and PLCL10MS stents after crimping and expansion.

Fig. S20.

DSA immediately, 7 day and 1 month after PLCL and PLCL10MS stents placement. The red dotted box highlights the position of the stents.

Fig. S21.

Reendothelializations after PLCL and PLCL10MS stents implantation for 7 day and 1 month in rabbit common carotid artery, shown in SEM images. Reendothelializations was completed after 1 months.

Fig. S22.

Images of PLCL and PLCL10MS stents segments of the common carotid artery stained with HE at 7 day and 1 month post-implantation.

Fig. S23.

Magnesium concentration in the blood of rabbits at 1 and 7 days post-implantation of PLCL and PLCL10MS stents. Sample size: n = 5. * P < 0.05, ** P < 0.01. NS, not significant.

Minor concerns:

1. S9 D: please specify the respective stainings on the figure?

[Reply]: Dear reviewer, thank you for your comment. We have specified the respective stainings on the figure (Figure S9) and modified the Figure legend as follows.

Fig.S9 *In vitro* cytotoxicity of PLCL loaded with different concentrations of MgSO₄.The cell viability of A) HUVECs, B) HASMCs and C) HT-22 cells were evaluated by CCK-8 assays, respectively. D) Live/dead cell staining (Calcein-AM (green), PI (red)) of HUVECs, HASMCs and HT-22 cells co-cultured with extract of PLCL loaded with different concentrations of MgSO₄, scale bars: 20 μm. Sample size: n=5.

2. Is there a specific reason for using human lines for endothelial cells and smooth muscle cells but mouse cell line for neurons (instead of say SH-SY5Y human cell line for neurons) ?

[Reply]: Dear reviewer, thank you for your comment. We do agree with you that both of HT-22 cells (Mouse Hippocampal Neuronal Cell Line) [X. Feng, K.A. Krogh, C.-Y. Wu, Y.-W. Lin, H.-C. Tsai, S.A. Thayer, L.-N. Wei, Receptor-interacting protein 140 attenuates endoplasmic reticulum stress in neurons and protects against cell death, Nature communications 5(1) (2014) 4487] and SH-SY5Y cells (Human Neuroblastoma Cell Line) [M. Pandey, V. Karmakar, A. Majie, M. Dwivedi, S. Md, B. Gorain, The SH-SY5Y cell line: a valuable tool for Parkinson's disease drug discovery, Expert Opinion on Drug Discovery 19(3) (2024) 303-316] can be used as cell line for neurons.

Based on your comment, we conducted a supplementary study by using SH-SY5Y cells under OGD/R. We added the description of “As a supplementary experiment, SH-SY5Y (a human neuronal cell line) cells were also treated under OGD/R and the cell viability was measured using the CCK-8 kit.” in the section of *In vitro* neuroprotection assessments in the revised manuscript.

We added the results of “According to cell viability assays, after OGD/R treatment, neural viability decreased, which was not mitigated by extracts of pure PLCL but was suppressed by extracts of PLCLxMS (Fig. 6B). And the results were very similar for the SH-SY5Y cell line in Fig. S12.” in the section of *In vitro* neuroprotection analysis using the OGD/R cell model in the revised manuscript.

Fig.S12.

In vitro neuroprotection analysis using the OGD/R cell model. The cell viability of SH-SY5Y cells were evaluated by CCK-8 assays. Sample size: n=5. * P< 0.05, *** P< 0.001 vs. OGD/R group. NS, not significant.

- Can the authors comment on why there is no additional burst of Mg^{2+} once the PLCL is hydrolysed to release the $MgSO_4$ particles from the inside of the stent?

[Reply]: Dear reviewer, thank you for your comment. We have added the description of “ As shown in Fig. 3D and Fig. S6, the $MgSO_4$ particles embedded in the stent surface were completely dissolved after 7 days of immersion, leaving behind cavities at their original locations. However, partly $MgSO_4$ particles remained deep inside the stent struts obtained from the cross-sectional morphology, this corresponding to the “Fast” stage. Because the rate of penetrating of water into the PLCL matrix is a fast and then slow process [41], after the surface layer of magnesium sulfate is dissolved, the shallow magnesium sulfate particles will come into contact with water molecules as the water is penetrating, thus dissolving and releasing magnesium ions. These magnesium ions are released into solution by outward diffusion. The rate of magnesium ion release also decreases during this phase as restricted by the rate of penetrating of water. The reported complete degradation time of PLCL stent in vivo is about 12 months, with a rapid degradation rate occurring by the sixth month [43], corresponding to the “Stable” stage. The dissolution and release of Mg^{2+} from the $MgSO_4$ particles within the stent may occur through a diffusion process in this study, leading to no additional burst release of Mg^{2+} , but instead a slow-release rate.” in the section of *in vitro* degradation analysis of the revised manuscript.

- The paper is underlining the role of Mg^{2+} ions in neuroprotection. The authors have also demonstrated that their material secretes only Mg^{2+} ions. However, in their previous study, Mg^{2+} alone has been shown to be insufficient to reduce infarct size (reference 6). It is not clear how the

released Mg^{2+} from their fabricated wires/stunt (at different concentrations) is able to produce contradicting results.

[Reply]: Dear reviewer, thank you for your comment. We apologize for the misunderstanding caused by the unclear description in the text. Both of this study and the previous research demonstrated the role of Mg^{2+} ions in neuroprotection. The result of present work agrees with the conclusion in reference 6.

The main purpose of our previous study (reference 6) is to verify the neuroprotective effects of intraperitoneally injected Mg alloy extracts on middle cerebral artery occluded mouse with reperfusion injury. Compared with the saline group, although the infarct volume was not obviously smaller in the Mg group, the mNSS scores and beam walking test scores of the MCAO rats in Mg group are also significantly different with the saline control group, suggesting the neuroprotective effects of Mg^{2+} .

Based on your comment, we added the description of “Based on the excellent *in vivo* neuroprotective effect of PLCL10MS, wires were further upgraded to spiral stents (Fig. S14A) and implanted in the CCA after MCAO (Fig. S14B). The PLCL10MS spiral stents still provided neuroprotection against I/R injury on the first day after implantation (Fig. S14C), further demonstrating the *in vivo* neuroprotective effects of PLCLxMS, not only in the form of wires but also spiral stents.” in the section of *Ex vivo* neuroprotection analysis and “After the implantation of PLCLxMS stents, the Mg^{2+} generated by the dissolution of $MgSO_4$ particles transferred to the distant ischemic area and cross the BBB into the brain, which contributed to neuron rescue, BBB protection, and enhanced CBF. Other reports suggested that injecting magnesium directly into target organs via proximal arteries may induce stronger neuroprotection [26].” in the section of Neuroprotection of Mg and treatment time window of ischemic stroke of the revised manuscript.

Response to Reviewer 2

Reviewer #2 (Remarks to the Author):

Current treatment options for ischemic stroke are limited to thrombolysis and thrombectomy, targeting reperfusion of ischemic tissue. However, ischemia-reperfusion injury per se may augment tissue injury and remains a fundamental problem that needs to be addressed. In their study Zhang and colleagues aim to introduce a Mg^{2+} eluting stent by incorporating neuroprotective $MgSO_4$ particles into a PLCL (substrate using 3D printing technology). They show that the release of Mg^{2+} provides sequential neuroprotection that aligns with the treatment window for acute ischemic stroke.

【Q1】 However, the advantage of release Mg^{2+} by the stand compared to the class IP or Iv injection is not clear. **【Q2】** Moreover, Magnesium neuroprotection in acute stroke in clinical set is still controversial, some clinical studies show no effect (i.e. DOI: 10.1056/NEJMoa1408827; DOI: 10.1161/STROKEAHA.118.021916), the authors do not refer to any of these studies.

[Reply to Q1]: Dear reviewer, thank you for your suggestion. Compared to the class IP or IV injection, release of magnesium ions by the stent is more targeted and effective, because of that the Mg^{2+} ions were directly delivered to the infarcted area via the cerebral artery by stent implantation, with a high utilization rate and better neuroprotective effect.

Therefore, we have added the description of “After the implantation of PLCLxMS stents, the Mg^{2+} generated by the dissolution of $MgSO_4$ particles transferred to the distant ischemic area and cross the BBB into the brain, contributing to neuron rescue, BBB protection, and enhanced CBF. Other reports suggested that injecting magnesium directly into target organs via proximal arteries might induce stronger neuroprotection [26].” in the section of Neuroprotection of Mg and treatment time window of ischemic stroke of the revised manuscript.

Moreover, we conducted a supplementary experiment to compare the neuroprotective effects of PLCL10MS implantation and Mg^{2+} intravenous injection. Our detail explanation is presented in your next question.

[Reply to Q2]: Dear reviewer, thank you for your suggestion. As for the controversy on Mg^{2+} neuroprotection in acute ischemic stroke patients, this might be attributed to the timing of administration. Besides, many patients who received magnesium experienced persistent vessel occlusion that was not recanalized before drug delivery, likely limiting the ability of magnesium and other agents to penetrate the target tissue. Furthermore, a significant subset of patients did not undergo either thrombectomy or thrombolysis; in the absence of primary reperfusion, the potential benefit of any neuroprotective agent alone is likely limited.

Therefore, we added a supplementary study to compare the neuroprotective effect of PLCL10MS in a permanent model with no reperfusion, cited your suggested references, and added the description of “Magnesium neuroprotection in acute stroke in clinical settings is controversial, possibly due to the timing of administration [68]. Recent studies have suggested that neuroprotectants should be tested in the reperfusion era, coinciding with the widespread application of mechanical thrombectomy [69]. The prehospital administration of $MgSO_4$ was deemed safe and enabled treatment initiation within two hours after stroke onset. However, it failed to show any improvement in clinical outcomes over a 90-day period in a randomized clinical trial [70]. One potential reason is that patients with AIS did not undergo thrombectomy or thrombolytic therapy; without initial reperfusion, the effectiveness of neuroprotective agents alone may be limited [71]. For comparison, the wire was also implanted into a permanent MCAO model (pMCAO) with no reperfusion, and it

was found that the neuroprotection provided by PLCL10MS in pMCAO was less effective than in tMCAO (Fig. S15), aligning with clinical study findings [70]. Consequently, reperfusion therapy (stent implantation) was combined with neuroprotective treatment (Mg^{2+} administration) by incorporating $MgSO_4$ particles into a PLCL stent to mitigate cerebral ischemia-reperfusion injury.” in the section of Neuroprotection of Mg and treatment time window of ischemic stroke in Discussion of revised manuscript.

Fig.S15.

Nissl staining and behavioral results of Group pMCAO, Group pMCAO+PLCL10MS and Group tMCAO+PLCL10MS. (A) Brain slices with Nissl staining and quantification of Nissl bodies; scale bars: 1mm and 25 μ m. (B) Results from the Longa score assessments, (C) rotarod test and (D and E) adhesive contact and removal test performance before stroke and up to 14 days post-stroke. Sample size: n = 6 for nissl staining; n = 12 for neurological tests. * P < 0.05, ** P < 0.01, *** P < 0.001, pMCAO+PLCL10MS vs. pMCAO group; # P < 0.05, ## P < 0.01, ### P < 0.001, tMCAO+PLCL10MS vs. pMCAO+PLCL10MS group.

More comments include:

1. This approach would have more impact if the PLCL10MS had bigger neuroprotective effect than Magnesium injected IP or IV as in previous studies, one more group (MCAO+stand+Mg²⁺ injected IP/IV) would have clarified the advantage of this technology.

[Reply]: Dear reviewer, thank you for your suggestion. We conducted a supplementary experiment to compare the neuroprotective effects of PLCL10MS implantation and Mg²⁺ intravenous injection. We found that these two treatment approaches had a similar neuroprotective trend, but the implantation of PLCL10MS exhibited better neuroprotective effects.

Therefore, we have added the supplementary experiment method of “ An additional group (intravenous injection, denoted as the IV group) was established by immersing the PLCL10MS stents in 10 ml of saline. The extracts were collected and replaced with fresh saline on days 1, 3, 5, and 7. At each time point, the extracts were administered to MCAO rats via the tail vein, after being disinfected and filtered. These rats were designated as the MCAO+Mg²⁺ from the PLCL10MS IV injection group. Some rats underwent 14 days of continuous behavioral testing, while others were sacrificed on day seven for brain removal and Nissl staining ” in the section of MCAO Model and Material Implantation in Animal Experiments of revised manuscript.

Moreover, we have added the description of “ A supplementary experiment was conducted to compare the the neuroprotective effects of PLCL10MS implantation and intravenous injection of Mg²⁺. It was found that these two treatment approaches had similar neuroprotective trends, but PLCL10MS exhibited superior neuroprotective effects (Fig. S24).” in the section of Neuroprotection of Mg and treatment time window of ischemic stroke in Discussion of revised manuscript.

Fig. S24.

Nissl staining and behavioral results of Group tMCAO+Mg²⁺ from PLCL10MS injected IV and Group tMCAO+PLCL10MS. (A) Brain slices with Nissl staining and quantification of Nissl bodies; scale bars: 1mm and 25 μ m. (B) Results from the Longa score assessments, (C) rotarod test and (D and E) adhesive contact and removal test performance before stroke and up to 14 days post-stroke. Sample size: n = 6 for nissl staining; n = 12 for neurological tests. * P < 0.05, ** P < 0.01, *** P < 0.001, tMCAO+PLCL10MS vs. tMCAO+Mg²⁺ from PLCL10MS injected IV group.

2. TTC staining is not ideal to quantify ischemia 7 days post-stroke, Nissle or H&E stainings are more appropriate.

[Reply]: Dear reviewer, thank you for your comment. We do agree with you that Nissl or H&E stainings are more appropriate to quantify ischemia 7 days post-stroke. We had conducted Nissl staining to evaluate the neuroprotective effects of our stents on 7 days post-stroke, and the related results were shown in Figure 7D.

Refer to reported protocol [F. Zhang, J. Chen, Infarct Measurement in Focal Cerebral Ischemia: TTC Staining, in: J. Chen, X.-M. Xu, Z.C. Xu, J.H. Zhang (Eds.), Animal Models of Acute Neurological Injuries II: Injury and Mechanistic Assessments, Volume 2, Humana Press, Totowa, NJ, 2012, pp. 93-98.], TTC staining is a supplementary methods to evaluate ischemic brain tissue[Nature Communications volume 14, Article number: 5984 (2023); Nature Communications volume 13, Article number: 6890 (2022); Nature Communications volume 11, Article number: 4078 (2020)]. Therefore, in order to provide more information, both staining methods were used in our study.

3. “In vivo” ROS detection is not really an in vivo quantification but ex-vivo. A different approach should be used to really measure the ROS levels in vivo. Same for “in vivo” Mg²⁺ distribution.

[Reply]: Dear reviewer, thank you for your suggestion. We have modified the related description and changed the “*in vivo*” into “*ex vivo*” in the revised manuscript.

Dear editor, we do agree with you that a real *in vivo* quantification of ROS levels and Mg²⁺ distribution is helpful. Most of the commercially available molecular probes are designed for usage at cellular level. Besides, due to the influence of laser stimulation, animal stress reaction, and the spontaneous fluorescence of peroxides in the blood, the *in vivo* quantification of ROS and Mg²⁺ may have a large range of errors. Therefore, we have added this limitation as “Therefore, it is crucial to regulate the release of Mg²⁺ to ensure they provide optimal neuroprotection within their limited loading capacity. Besides, one limitation is that the quantification of ROS levels and Mg²⁺ distribution was analyzed by *ex vivo* but not *in vivo* approach. With the development of molecular probe technology, *in situ* evaluation of reactive species in rat brain should be conducted.” in the section of The time-match between stroke treatment time-window and Mg²⁺ release in Discussion of the revised manuscript.

4. Most of the behavioral seem to have a significant difference only at 14 days in between groups, however the authors hypothesized that stands might have acute and long term functional recovery. Why are the PLCL5MS and PLCL10MS only beneficial after 2 weeks if ischemic volume, BBB leakage etc are already neuroprotective at 1 week?

[Reply]: Dear reviewer, thank you for your comment. We apologize for the misunderstanding caused by the unclear description in the text.

Compared with the tMCAO groups, the PLCL5MS and PLCL10MS groups exhibited better neurological function recovery from day 1 to day 14. We indicated these significant differences in the Figure 8A, C, E and F, and modified the figure legend in the revised manuscript as “All the behavioral test results indicated that rats in the tMCAO+PLCL5MS group exhibited better mobility than those in the tMCAO group from day 1 to day 14, although their performance was not as good as that of the tMCAO+PLCL10MS group. These results suggest that both PLCL5MS and PLCL10MS provided significant neuroprotection, which was evident from day 1 through day 14.” .

Fig. 8. Behavioral evaluation of tMCAO rat models implanted with PLCLxMS wires after ischemia/reperfusion injury. (A) Longa score assessments. (B and C) Results from the Rotarod test. (D-F) Adhesive contact and removal test performance before stroke and up to 14 days post-stroke. (G) Open field test conducted 7 days post-stroke. (H) Walking distance and time in the open field. (I) Magnesium concentration in the blood and ipsilateral infarction brain (IIB) of rats at 1 and 7 days post-implantation of PLCLxMS wires. Sample size: $n = 12$ for neurological tests; $n = 3$ for magnesium content measurement tests. ** $P < 0.01$, *** $P < 0.001$, tMCAO+PLCL5MS or tMCAO+PLCL10MS vs. tMCAO group; # $P < 0.05$, ## $P < 0.01$, ### $P < 0.001$, tMCAO+PLCL5MS or tMCAO+PLCL10MS vs. tMCAO+PLCL group; § $P < 0.05$, §§ $P < 0.01$, §§§ $P < 0.001$, tMCAO+PLCL10MS vs. tMCAO+PLCL5MS.

5. The authors also hypothesized that stents may offer sequential neuroprotection to remote ischemic brain tissues, however no mechanisms are shown, no trigger is hypothesized for remote neuroprotection.

[Reply]: Dear reviewer, thank you for your question. We have modified our description as “ The staged-release of Mg^{2+} is supposed to provide sequential neuroprotection that aligns with the treatment window for acute ischemic stroke.” in the section of Abstract.

As for the process of transferring Mg^{2+} to remote ischemic brain tissues, we have added the description of “After the implantation of PLCLxMS stents, the Mg^{2+} generated by the dissolution of $MgSO_4$ particles transferred to the distant ischemic area and cross the BBB into the brain, which contributed to neuron rescue, BBB protection, and enhanced CBF. Other reports suggested that injecting magnesium directly into target organs via proximal arteries may induce stronger neuroprotection [26].” in the section of Neuroprotection of Mg and treatment time window of ischemic stroke of the revised manuscript.

Our PLCLxMS stent is designed to provide potential sequential neuroprotection, and this hypothesis is made on the basis of staged release of Mg^{2+} (Fig.3) and animal study (Fig.7 and Fig.8). The detailed description was presented in the section of The time-match between stroke treatment time-window and Mg^{2+} release in Discussion of the revised manuscript.

Based on your comment, in order to better illustrate this hypothesis, we have added the description of “The proposed neuroprotection mechanism of Mg on each stage after the stroke incidence was summarized in Table S1.” in Discussion of the revised manuscript.

Table S1. The proposed sequential neuroprotection mechanism of the stent

Stage	Time after stroke incidence	Physiological process of reperfusion injury	Neuroprotection of Mg^{2+}	Reference
Burst	Seconds to hours in rodent stroke models	BBB becomes dysregulated; ROS or DNA damage mediated neuronal apoptosis; Ca^{2+} and glutamate mediated excitotoxicity; inflammatory responses mediated secondary injuries	Inhibit the release of glutamate; inhibit Ca^{2+} overload; improve post-ischemic vascular perfusion; reduce ROS production and mitochondrial damage	[17, 20, 24, 65-67]
Fast	The first 3 weeks of the initial injury in rodent models of stroke	Inflammatory response peaks; the infarct volume ceases to expand; astrocytes proliferate locally to create a physical barrier; pro-repair mechanisms up-regulate; BBB repair;	Promote proliferation of astrocytes and neurogenesis; reduce ROS production and mitochondrial damage caused by inflammation	[73-75]
Stable	After Fast stage	Persistent inflammatory responses and new vessels form; Neural circuit plasticity diminish	Mg^{2+} concentration reaches dynamic equilibrium	[17]

6. Number of animals per group in Fig 7 is only showed in panel c.

[Reply]: Dear reviewer, thank you for your suggestion. We have modified all the figures and added the animal numbers accordingly.

7. How was the sample size measured?

[Reply]: Dear reviewer, thank you for your question. The sample size is determined on 3Rs (Replace, Reduce and Refine) principle for animal experiments, and is kept to a minimum number while obtaining statistically significant data.

The sample size of the brain integrity and cerebral blood flow data in Figure 7 and the sample size of the ex vivo Mg^{2+} , ROS, and Ca^{2+} concentration data in Figure 8 and supplementary figures were controlled as $n = 3-6$. Due to the high variability of the behavioral data in Figure 8, the sample size was expanded to 12.

8. Some statistics should use two-ways Anova.

[Reply]: Dear reviewer, thank you for your suggestion. We have conducted the two-ways Anova analysis on the data, pertaining to various time points and groups, in Figure 3C, Figure 7J and Figure 8A, C, E and F. The P-value results were labeled in the figures. The two-ways ANOVA analysis was also added into the Method section in the revised manuscript.

We have modified our description of “All results were expressed as mean \pm standard deviation. SPSS 17.0 was used for statistical analysis. Differences between two groups were compared using the two-tailed unpaired Student's t test and differences in means among multiple groups were analyzed using one-way ANOVA followed by the Tukey/Games-Howell post hoc correction. Data pertaining to various time points and groups were subjected to the two-ways ANOVA analysis, which was also followed by the Bonferroni post hoc correction.” in the revised manuscript.

9. In Methods section, laser speckle imaging acquisition and analysis is not included. Moreover, because carbon dioxide (CO_2) state might affect the results, a systemic arterial blood-gas measurement needs to be performed at the end of each experiment to record pCO_2 and blood pressure (magnesium could reduce blood pressure).

[Reply]: Dear reviewer, thank you for your suggestion. The laser speckle imaging acquisition and analysis method have been added in the section of Measurement of cerebral blood flow in Animal Experiments of the revised manuscript as follows: “Relative cerebral blood flow (CBF) was determined using laser speckle imaging (LSI, RWD Science Co., China). The rats, anesthetized with 1%-3% enflurane after initial 5% enflurane administration, had their cranial skin shaved and disinfected with iodophor. The animal was then positioned in a prone posture with its head secured in a stereotaxic device. A longitudinal incision was made in the cranial skin. The skull was carefully thinned from bregma to lambda under a dissecting microscope using a dental drill until the LSI could clearly detect CBF. CBF was measured for at least five minutes in each hemisphere of the brain during the experiment. Color bands representing perfusion units were used to visualize CBF, and the average CBF perfusion units for rats in both hemispheres were calculated. The CBF on the damaged side was expressed as a percentage of the value on the contralateral side.”

According to your comments, we added the description of “At the end of each laser speckle

imaging experiment, the pCO₂ levels and blood pressure of all rats were recorded, with minimal variations (Fig. S13), consistent with observations made previously through direct intra-arterial infusion of MgSO₄ [58].” in the section of *Ex vivo* neuroprotection analysis in the revised manuscript.

Fig. S13.

Measurement of (A) pCO₂ and (B) mean arterial blood pressure at the end of each experiment of LSI. Sample size: n=6.

10. **【Q1】** To mimic a post-thrombotic event with reperfusion damage, a clot-model would be more appropriate than the filament model. **【Q2】** Comparing the effect of PLCL10MS in a permanent model with no reperfusion would address the effect on the reperfusion injury.

[Reply to Q1]: Dear reviewer, thank you for your suggestion. We do agree with you that a clot-model is amenable to thrombolytic therapies with reperfusion damage. The infarction produced by this model is smaller and more variable than the filament model due to the continuous process of endogenous thrombolysis [Liu, F., McCullough, L.D. (2014). The Middle Cerebral Artery Occlusion Model of Transient Focal Cerebral Ischemia. In: Milner, R. (eds) Cerebral Angiogenesis. Methods in Molecular Biology, vol 1135. Humana Press, New York, NY].

The clinical background of our research is that, for patients with vascular occlusion caused by atherosclerosis of large vessels, stent placement after thrombectomy is often necessary for complete revascularization. For these patients, we want to endow the stent with additional neuroprotective effects to combat ischemia-reperfusion injury. Therefore, we used a filament model to mimic mechanical thrombectomy.

[Reply to Q2]: Dear reviewer, thank you for your suggestion. We added a supplementary study to compare the neuroprotective effect of PLCL10MS in a permanent model with no reperfusion.

Therefore, we have added the description of “Male SD rats, approximately 300g each, were divided into five groups. One group served as the healthy, sham-operated control, while others underwent the MCAO procedure with or without implantation of PLCL, PLCL5MS, and PLCL10MS wires (tMCAO+PLCL, tMCAO+PLCL5MS, and tMCAO+PLCL10MS groups). Three days prior to this procedure, rats received Aspirin (10 mg/kg/day) and Clopidogrel (7.5 mg/kg/day) until sacrifice. The MCAO procedure was performed as previously described [101]. Rats were anesthetized with 1-3% isoflurane, followed by right MCA occlusion using a filament. The filament was withdrawn after 2 hours to allow reperfusion of the ischemic hemisphere. For the pMCAO group, the filament was not withdrawn. Then, 1 cm long PLCLxMS wires were inserted through an opening in the ECA into the CCA.” in the section of MCAO Model and Material Implantation in the revised manuscript.

And we have added the description of “To address the effect of PLCLxMS on the reperfusion injury, the wire was also implanted into a permanent MCAO model (pMCAO) with no reperfusion for comparison and it was found that the neuroprotection of PLCL10MS in pMCAO is less effective than that in tMCAO (Fig. S15), which is consistent with findings in clinical studies.” in the revised manuscript.

Fig. S15.

Nissl staining and behavioral results of Group pMCAO, Group pMCAO+PLCL10MS and Group

tMCAO+PLCL10MS. (A) Brain slices with Nissl staining and quantification of Nissl bodies; scale bars: 1mm and 25 μ m. (B) Results from the Longa score assessments, (C) rotarod test and (D and E) adhesive contact and removal test performance before stroke and up to 14 days post-stroke. Sample size: n = 6 for nissl staining; n = 12 for neurological tests. * P < 0.05, ** P < 0.01, *** P < 0.001, pMCAO+PLCL10MS vs. pMCAO group; # P < 0.05, ## P < 0.01, ### P < 0.001, tMCAO+PLCL10MS vs. pMCAO+PLCL10MS group.

Response to Reviewer 3

Reviewer #3 (Remarks to the Author):

This is a study of a novel Mg-eluting PLCL stent for ischemia-reperfusion neuroprotection with an evaluation of its mechanical, surface, and structural properties, elution characteristics, and in vitro and in vivo biologic effects. The authors characterized the novel biodegradable stent and demonstrated its potential utility in vitro and in vivo. There are a few queries that require consideration:

1. There are minor grammatical errors sprinkled throughout the manuscript including “This stage is characterized the penetrating of water into the PLCL matrix and the outward diffusion of Mg^{2+} from within the stent”, and “And the PLCLxMS sheets exhibited a significant increase...”. Also, CAD is used both as an abbreviation for computer-aided design as well as coronary artery disease.

[Reply]: Dear reviewer, thank you for your comment. We apologize for the grammatical errors of our manuscript. We have now worked on both language and readability and have also involved native English speakers for language corrections.

We have modified our description as “This stage **features in** the penetrating of water into the PLCL matrix and the outward diffusion of Mg^{2+} from within the stent” and “And **the mechanical properties of** the PLCLxMS sheets exhibited a significant increase in tensile modulus with the increase in $MgSO_4$ content (Fig. 4C), which is consistent with the Halpin-Tsai theoretical model for estimating the modulus of nanofiller-reinforced composites” in the revised manuscript. And we also modified our description and CAD is only used as an abbreviation for computer-aided design.

2. **【Q1】** In the X-ray diffractometer data in Figure 2D, can the authors speculate why there is a $MgSO_4$ (101) peak for the control (PLCL group)? **【Q2】** How was the potential for Mg contamination on PLCL alone stents controlled if using the same 3D extrusion deposition printer?

[Reply to Q1]: Dear reviewer, thank you for your comment. We apologize for the misunderstanding caused by the unclear demonstration of Fig.2D in the text. The $MgSO_4$ (101) peak was located at about $2\theta = 22.7^\circ$, and the disputable peak for PLCL was located at about $2\theta = 23.0^\circ$.

Based on your comment, we have modified Figure 2D, and the characteristic $MgSO_4$ XRD peaks were indicated by the red dash lines.

[Reply to Q2]: Dear editor, thank you for your comment. The possibility of Mg contamination on

PLCL alone stents is low. When printing different PLCLxMS materials, we clipped out the remaining material and wiped the bin with anhydrous ethanol to prevent contamination between batches.

3. As per Figure S6, surface roughness increases due to magnesium dissolution within 7 days. Can the authors speculate what this early and progressive roughness will do to the thrombogenic potential of the stent over time?

[Reply]: Dear reviewer, thank you for your comment. We have supplemented the hemocompatibility experiments (platelet adhesion test and hemolysis analysis) of PLCL, PLCL5MS, and PLCL10MS before and after 7 d of immersion (Fig. R7).

Therefore, we have added the description of “Fresh blood from New Zealand rabbits was centrifuged at 1000 rpm for 15 minutes to obtain platelet-rich plasma (PRP) from the supernatant. PLCL, PLCL5MS, and PLCL10MS sheets, both before and after immersion in PBS for 7 days and drying (designated as PLCL-7D, PLCL5MS-7D, and PLCL10MS-7D) were placed in a 24-well plate. 0.4 mL of PRP was added to each well to ensure the samples were submerged. After 60 minutes of incubation at 39°C, the PRP was removed, and the discs were gently rinsed three times with PBS. Platelets on the sheet surfaces were fixed using a 2.5% glutaraldehyde solution for 60 minutes, then dehydrated in gradient ethanol solutions (50%, 60%, 70%, 80%, 90%, 100%) for 10 minutes each. The sheets were then dried at 25°C. The morphology of the platelets on the disc surfaces was observed using SEM. Three parallel samples were prepared for each material.” in the added section of Platelets adhesion test and “ Fresh blood was collected from healthy New Zealand rabbits, and 3.8 wt. % sodium citrate was added to the blood samples. The blood-to-citrate volume ratio was 9:1. The blood samples, mixed with sodium citrate, were diluted with PBS at a 4:5 blood-to-PBS ratio. The diluted blood samples were transferred into a centrifuge tube and immersed in 10 ml PBS at 39 °C for 30 minutes. Deionized water and PBS were used as the positive and negative control groups, respectively. The sheets were immersed in 10 mL PBS at 39°C for 30 minutes, followed by the addition of 0.2 mL of blood. These mixtures were incubated in the alloy extracts at 39 °C for 60 minutes. The samples were then centrifuged, and the absorbance of the supernatant at 545 nm was measured using an enzyme-plate analyzer (Bio-RAD 680). Three parallel samples were prepared for each material. ” in the added section of Hemolysis rate analysis in the added section of Platelets adhesion test in the revised manuscript.

And we have added the description of “ According to the platelet adhesion test, no obvious platelet aggregations were observed on the surface of the PLCL, PLCL5MS, and PLCL10MS before and after immersion for 7 days (Fig. S10A), indicating that the platelets were not activated [57]. The hemolysis rates of all materials were less than 5%, classifying them as blood-compatible materials according to the ISO 10993-4 standard (Fig. S10B). These results suggest that the dissolution of MgSO₄ and the surface morphology changes in PLCL5MS and PLCL10MS had minimal effects on their hemocompatibility. ” in the section of *In vitro* biocompatibility test of the revised manuscript.

Fig. S10.

(A) Scanning electron microscope images of platelet adhesion on the surface of PLCL, PLCL5MS, and PLCL10MS before and after immersion for 7d; (B) Rate of hemolysis. (n = 3, independent samples). Sample size: n = 3 for hemolysis test. * P < 0.05, ** P < 0.01.

4. 【Q1】 It is interesting that in the *in vitro* experiments, Mg continued to be released up to 56 days. However, in the *in vivo* studies, Mg concentration reduced to normal levels by day 7. Is the reduction of Mg conc in the IIB by day 7 related to the completion of Mg dissolution, or could endothelialization of the stent surface play a role here? 【Q2】 In other words, could endothelialization (which can take place over a similar timeframe in rats) affect the dissolution of Mg from the stent and subsequent effectiveness since it may prevent the Mg from being released and reaching downstream targets? Did the authors histologically evaluate the spiral stent explants for endothelialization after 7 days.

[Reply to Q1]: Dear reviewer, thank you for your comment. We apologize for the misunderstanding caused by the unclear description of the results in Fig.8I and Fig.S12. In the *in vitro* experiments (Fig.3B), the release of Mg²⁺ mainly occurs in 0~7 days. The reduction of Mg concentration in the IIB by day 7 mainly ascribed to the the recovery of BBB integrity in MCAO rat by day 7, which

enables the dynamic equilibrium of Mg^{2+} concentrations in brain.

Therefore, we have modified our description as “As depicted in Fig. 8I, Mg^{2+} concentrations in the blood of rats from the experimental group (tMCAO+ PLCL10MS and tMCAO+PLCL5MS) were higher than those in the control group (sham, tMCAO, and tMCAO+PLCL) on days 1 and 7. Mg^{2+} concentrations in the ipsilateral infarction brain (IIB) of rats in the experimental group were also higher on day 1 but showed no differences by day 7. Moreover, Mg^{2+} levels in the infarction contralateral brain (ICB) at 1 and 7 days post-implantation of PLCLxMS wires remained similar to those of the control groups (Fig. S16). Endothelialization on the implanted wires was not observed by day 7 (Fig. S17). One possible explanation is that the recovery of BBB integrity in tMCAO rats by day 7 facilitated the dynamic equilibrium of Mg^{2+} concentrations in the brain [59].” in the section of *Ex vivo* Mg^{2+} , ROS and Ca^{2+} concentration detection of the revised manuscript.

[Reply to Q2] : Dear reviewer, thank you for your comment. According to your comment we conducted two supplementary studies.

Supplementary study 1: The surface morphology of the PLCL, PLCL5MS and PLCL10MS wires after implantation for 7 days were observed. And we added the description of “Endothelialization on the implanted wires was not observed by day 7 (Fig. S17). One possible explanation is that the recovery of BBB integrity in tMCAO rats by day 7 facilitated the dynamic equilibrium of Mg^{2+} concentrations in the brain [59]” in the section of *Ex vivo* Mg^{2+} , ROS and Ca^{2+} concentration detection of the revised manuscript.

Fig. S17.

Scanning electron microscope images of the PLCL, PLCL5MS and PLCL10MS wires after implantation for 7 days.

Supplementary study 2: We implanted the stent into the CCA of healthy rabbit models to study

the structural integrity and the in vivo Mg^{2+} release of the opened stent.

Therefore, we have added the method description of “Forty male New Zealand rabbits weighing between 3 and 3.5 kg were utilized in the investigation; ten served as the healthy, sham-operated control, fifteen were successfully implanted with PLCL stents and fifteen with PLCL10MS stents. The stents were sterilized by ethylene oxide. All experimental procedures received approval from the Capital Medical University's Institutional Animal Investigation Committee and were conducted in accordance with the National Institutes of Health's standards for the Care and Use of Laboratory Animals. During implantation, a PLCL or PLCL10MS stent, was deployed in the common carotid artery (CCA) of New Zealand rabbits. To do this, after 12 h of fasting, anesthesia was performed by intravenous injection of 1 ml/kg pentobarbital. Then the rabbit was fixed in a supine position and the stent was delivered to CCA via minimally invasive surgery from femoral artery. The stents were deployed with the recommended pressure of 12~16 atm for PLCL and PLCL10MS stents. Digital Subtraction Angiography (DSA) were conducted immediately after the balloon deflation to confirm the patency of the blood flow and the correct placement of the stent. The arteries contained the stents at each time point (7 days and 1 month) were harvested for subsequent analysis. And some samples were embedded in paraffin and sliced into 5 μm sections, and then stained with hematoxylin-eosin (HE). Others were fixed with 2.5% glutaraldehyde and left for 12 h. Subsequently these were dehydrated using a series of gradient ethanol solutions of 30%, 50%, 70%, 80%, 90%, 100%, and 100% with each step lasting 15 min. The targeted artery segments were then longitudinally bisected coated with a thin layer of gold before examined using a scanning electron microscope (SEM). Blood from rabbit CCA near the stent implantation position was collected on days 1 and 7 post-implantation and Mg concentration was determined via ICP-OES (iCAP6300, Thermo) after digested with HNO_3 .” in the section of PLCL10MS stent implantation in rabbit CCA model and analysis in the revised manuscript.

We added the results of “The optical morphology of the PLCL and PLCL10MS stents during crimp and expansion are shown in Fig. S19. Digital Subtraction Angiography images (DSA) in Fig. S20 show that both of the PLCL and the PLCL10MS stents were successfully implanted and no stenosis were observed at 7 days and 1 month. Combined with the SEM (Fig. S21) and HE staining (Fig. S22) results, the PLCL and PLCL10MS stents struts in rabbits CCA were partly covered by the neointima at 7 days and were fully embedded by intima at 1 month. The concentration of magnesium at 1 day in CCA blood of PLCL10MS stent group was higher than that of the sham and PLCL stent group, but with no statistical difference at 7 day (Fig. S23).” in the section of Implantation result and histological analysis of PLCL10MS stent in rabbit CCA model in the revised manuscript.

Fig. S19.

Optical images of the PLCL and PLCL10MS stents after crimping and expansion.

Fig. S20.

DSA immediately, 7 day and 1 month after PLCL and PLCL10MS stents placement. The red dotted box highlights the position of the stents.

Fig. S21.

Reendothelializations after PLCL and PLCL10MS stents implantation for 7 day and 1 month in rabbit common carotid artery, shown in SEM images. Reendothelializations was completed after 1 months.

Fig. S22.

Images of PLCL and PLCL10MS stents segments of the common carotid artery stained with HE at 7 day and 1 month post-implantation.

Fig. S23.

Magnesium concentration in the blood of rabbits at 1 and 7 days post-implantation of PLCL and PLCL10MS stents. Sample size: n = 5. * P< 0.05, ** P< 0.01. NS, not significant.

- The authors provide a reference for the neuroprotective effect of Mg being dose dependent, although there was no dose dependent effect on in vitro neuroprotection shown in vitro beyond 10% MgSO₄. How can the authors reconcile this?

[Reply]: Dear reviewer, thank you for your comment. We have replaced “dose-dependent” as “dose-related” accordingly in the revised manuscript.

And based on your comment, we have modified our description as “ Among different samples, PLCL10MS and higher MgSO₄ contents exhibited more significant effects on viability rescue and inhibition of Ca²⁺ influx, ROS production, and cell apoptosis, though no dose-related effect was observed beyond 10% MgSO₄ (Fig. 6B-H). This suggested that the PLCL10MS provided better neuroprotective effects on OGD/R cells than that of PLCL5MS.” in the section of *In vitro* neuroprotection analysis using the OGD/R cell model.

- The authors write “As shown in Fig. 3D and Fig S6, the MgSO₄ particles embedded in the stent surface were completely dissolved after 7 days of immersion, leaving behind cavities at their original locations. However, MgSO₄ particles remained inside the stent struts obtained from the cross-sectional morphology.” Since Mg levels in the IIB normalized by day 7, one could argue that this was all surface Mg and the inside Mg was not required to produce an effect in vivo. What is the importance, then, of the internal Mg as opposed to simply a stent surface coating of Mg?

[Reply]: Dear reviewer, thank you for your comment.

As for the release of Mg ions, we apologize for the misunderstanding caused by the unclear

description in the text. We have modified the description of “ As shown in Fig. 3D and Fig S6, the MgSO₄ particles embedded in the stent surface were completely dissolved after 7 days of immersion, leaving behind cavities at their original locations. However, partly MgSO₄ particles remained deep inside the stent struts obtained from the cross-sectional morphology, this corresponding to the “Fast” stage. Because the rate of penetrating of water into the PLCL matrix is a fast and then slow process[41], after the surface layer of magnesium sulfate is dissolved, the shallow magnesium sulfate particles will come into contact with water molecules as the water is penetrating, thus dissolving and releasing magnesium ions. These magnesium ions are released into solution by outward diffusion. The rate of magnesium ion release also decreases during this phase as restricted by the rate of penetrating of water. The reported complete degradation time of PLCL stent *in vivo* is about 12 months, with a rapid degradation rate occurring by the sixth month [43], corresponding to the “Stable” stage. The dissolution and release of Mg²⁺ from the MgSO₄ particles within the stent may occur through a diffusion process in this study, leading to no additional burst release of Mg²⁺, but instead a slow release rate.” in the section of *in vitro* degradation analysis of the revised manuscript.

The interal MgSO₄ particles have two roles.

The first one is that they are functionalized as mechanical reinforcement fillers, as demonstrated in Fig.4 and Fig.5.

The second one is that the sustained release of Mg ions with 7 days (corresponding to the “Fast” stage) are supposed to provide further neuroprotections, and we have added the description of “The proposed neuroprotection mechanism of Mg on each stage after the stroke incidence was summarized in Table S1.” in Discussion of the revised manuscript.

Table S1. The proposed sequential neuroprotection mechanism of the stent

Stage	Time after stroke incidence	Physiological process of reperfusion injury	Neuroprotection of Mg ²⁺	Reference
Burst	Seconds to hours in rodent stroke models	BBB becomes dysregulated; ROS or DNA damage mediated neuronal apoptosis; Ca ²⁺ and glutamate mediated excitotoxicity; inflammatory responses mediated secondary injuries	Inhibit the release of glutamate; inhibit Ca ²⁺ overload; improve post-ischemic vascular perfusion; reduce ROS production and mitochondrial damage	[17, 20, 24, 65-67]
Fast	The first 3 weeks of the initial injury in rodent models of stroke	Inflammatory response peaks; the infarct volume ceases to expand; astrocytes proliferate locally to create a physical barrier; pro-repair mechanisms up-regulate; BBB repair;	Promote proliferation of astrocytes and neurogenesis; reduce ROS production and mitochondrial damage caused by inflammation	[73-75]
Stable	After Fast stage	Persistent inflammatory responses and new vessels form; Neural circuit plasticity diminish	Mg ²⁺ concentration reaches dynamic equilibrium	[17]

7. Although numerous studies were completed on PLCLxMS stent degradation, including its effect on the mechanical properties of the stent, this was performed in the context of Mg release and did not address PLCL stent degradation over the longer term. The full time course of PLCL degradation was not addressed but should be mentioned or referenced.

[Reply]: Dear reviewer, thank you for your comment. We have added the description of “ As shown in Fig. 3D and Fig S6, the MgSO₄ particles embedded in the stent surface were completely dissolved after 7 days of immersion, leaving behind cavities at their original locations. However, partly MgSO₄ particles remained deep inside the stent struts obtained from the cross-sectional morphology, this corresponding to the “Fast” stage. Because the rate of penetrating of water into the PLCL matrix is a fast and then slow process[41], after the surface layer of magnesium sulfate is dissolved, the shallow magnesium sulfate particles will come into contact with water molecules as the water is penetrating, thus dissolving and releasing magnesium ions. These magnesium ions are released into solution by outward diffusion. The rate of magnesium ion release also decreases during this phase as restricted by the rate of penetrating of water. The reported complete degradation time of PLCL stent in vivo is about 12 months, with a rapid degradation rate occurring by the sixth month [43], corresponding to the “Stable” stage. The dissolution and release of Mg²⁺ from the MgSO₄ particles within the stent may occur through a diffusion process in this study, leading to no additional burst release of Mg²⁺, but instead a slow release rate.” in the section of *in vitro* degradation analysis of the revised manuscript.

8. N values for number of animals and in vitro studies should be included.

[Reply]: Dear reviewer, thank you for your comment. N values for number of animals and *in vitro* studies have been included accordingly as follows:

Fig. 3 ((B) Staged-cumulative release curve of Mg²⁺ at different stages. n=3),

Fig. 4 ((B) Stress-strain curve of PLCLxMS sheets. n=3. ... (F) Radial force with compression of PLCLxMS stents. n=3.),

Fig. 5 ((D) Hardness of the PLCL surface after immersion in various solutions. n=3.),

Fig. 6(... (H) Quantification of apoptosis rate from (G). n=5. * P< 0.05, ** P< 0.01, *** P< 0.001 vs. OGD/R group. NS, not significant.).

Fig. 7(... (J) Comparison of relative CBF across groups. Sample size: n = 6. * P< 0.05, ** P< 0.01...

Fig. 8(... (I) Magnesium concentration in the blood and ipsilateral infarction brain (IIB) of rats at 1 and 7 days post-implantation of PLCLxMS wires. Sample size: n = 12 for neurological tests; n = 3 for magnesium content measurement tests. ...

REVIEWER COMMENTS

Reviewer #2 (Remarks to the Author):

The authors have addressed my questions

[Reply]: Dear reviewer, thank you for your comment.

Reviewer #3 (Remarks to the Author):

I commend the authors for their additional studies, responses, and the edits made to address comments made in the first manuscript review. Specifically, all of my comments/concerns have been adequately addressed in the revised version of this manuscript.

[Reply]: Dear reviewer, thank you for your comment.

Reviewer #4 (Remarks to the Author):

This paper explores an important topic in the treatment of acute ischemic stroke, particularly the therapeutic challenges encountered by patients with large - vessel occlusion after thrombectomy. In the work, the prepared PLCL scaffolds with different concentrations of MgSO₄ using 3D - printing technology, aiming to explore their potential application in alleviating ischemia - reperfusion injury.

Overall, this study provides one potential way for the later - stage treatment of stroke by evaluating the performance of the scaffold, the release mechanism of magnesium ions, and the neuroprotective effects.

However, the some key points still require the further resolution:

1. Please clearly explain the related clinical pathological mechanism of acute ischemia stroke to support the rationality of the staged-release of Mg²⁺ in time and dose, and further state the clinical application significance that comes with the staged-release.

[Reply]: Dear reviewer, thank you for your comment. We have added related discussion and modified Table S1 as follows:“ In this study, the proposed time match between the release of Mg^{2+} and the treatment windows for ischemic stroke were illustrated in Fig. 9 and Table S3[17, 20, 24, 73-75, 81-83]. During the "Burst" stage, substantial amounts of Mg^{2+} ions are released, exerting neuroprotective effects in the hyperacute stage of AIS treatment. The subsequent "Fast" stage features sustained Mg^{2+} release, which facilitates neural repair and attenuates inflammatory responses, ultimately reaching dynamic equilibrium in the "Stable" stage.

The hyperacute stage, representing the optimal therapeutic window for AIS [17], is characterized by ROS-mediated neuronal apoptosis and DNA damage. Excitotoxicity occurs through Ca^{2+} accumulation and glutamate-induced overstimulation of NMDA receptors. These primary injuries are compounded by ROS-specific inflammatory responses, leading to secondary damage. The released Mg^{2+} ions function as endogenous antagonists of voltage- and ligand-gated calcium channels, including NMDA receptors, thereby reducing ROS production and mitochondrial damage. This mechanism effectively suppresses ischemia-reperfusion injury, as demonstrated in Fig. 7. During the acute stage, while inflammatory responses peak, the rate of cerebral infarction volume expansion shows significant reduction [81]. Brain-resident microglia serve as the primary responders to injury, releasing pro-inflammatory cytokines that disrupt the BBB and exacerbate inflammation [82]. This initial response is subsequently accompanied by ROS-mediated inflammatory cascades [83]. Concurrently, astrocytes proliferate to establish a protective barrier. The Mg^{2+} released during the "Fast" stage inhibits mitochondrial ROS production and regulates hippocampal neural stem cell proliferation [84], thereby promoting neural repair in rats. During the subacute stage, which corresponds to the Stable phase, most reperfusion-induced neural damage is repaired. Meanwhile, magnesium sulfate particles retained within the stent struts maintain the mechanical integrity of the stent through their slow dissolution process. Excitingly, PLCLxMS stents also exhibited better mechanical performance than pure PLCL stents, as shown

in Fig. 5A.” in the section of The time-match between stroke treatment time-window and Mg²⁺ release in the revised manuscript.

Table S3. The proposed sequential neuroprotection mechanism of the stent

Stages of ischaemia	Time	Physiological mechanism	Stages of Mg ²⁺ release	Neuroprotection of Mg ²⁺	Reference
Hyperacute	Seconds to hours in rodent stroke models	BBB becomes dysregulated; ROS or DNA damage mediated neuronal apoptosis; Ca ²⁺ and glutamate mediated excitotoxicity; inflammatory responses mediated secondary injuries	Burst	Inhibit the release of glutamate; inhibit Ca ²⁺ overload; improve post-ischemic vascular perfusion; reduce ROS production and mitochondrial damage	[17, 20, 24, 73-75]
Acute	The first 3 weeks of the initial injury in rodent models of stroke	Inflammatory response peaks; the infarct volume ceases to expand; astrocytes proliferate locally to create a physical barrier; pro-repair mechanisms up-regulate;	Fast	Promote proliferation of astrocytes and neurogenesis; reduce ROS production and mitochondrial damage caused by inflammation	[81-83]
Subacute	After Fast stage	Persistent inflammatory responses and new vessels form; Neural circuit plasticity diminish	Stable	Mg ²⁺ concentration reaches dynamic equilibrium	[17]

For further state the clinical application significance that comes with the staged-release. We have added the discussion as follows:“ In AIS treatment, the staged release of Mg²⁺ should align with the therapeutic time window, given the time-sensitive nature of stroke and the dose-dependent neuroprotective properties of Mg²⁺ [80]. Ghozy et al. demonstrated that ultra-early administration of neuroprotective agents is essential for achieving therapeutic efficacy [85]. This is supported by Marinov et al.'s findings, which revealed that MCAO animal models treated with 90 mg/kg MgSO₄ showed

significantly greater infarct volume reduction compared to the 30 mg/kg group [25]. These findings suggest that optimizing Mg^{2+} release during the hyperacute stage could potentially enhance the neuroprotective efficacy of the PLCLxMS stent. Beyond the hyperacute stage, timely Mg^{2+} delivery plays a crucial role in mitigating apoptosis and necrosis, preserving neurological function, and promoting the recovery of compromised brain tissue. Furthermore, maintaining optimal stent mechanical integrity throughout the subacute stage is critical for minimizing stent restenosis rates and ensuring long-term therapeutic outcomes. One limitation is that the quantification of ROS levels and Mg^{2+} distribution was analyzed *ex vivo*, rather than using an *in vivo* approach. With the development of molecular probe technology, *in situ* evaluation of reactive species in the rat brain should be conducted.” in the section of The time-match between stroke treatment time-window and Mg^{2+} release in the revised manuscript.

2.Lines 101 - 140: please introduce the advantages or the innovation of PLCL. The used When introducing the properties of experimental materials, some comparative analysis with other commonly used materials can be added to highlight the advantages and characteristics of the materials in this study. For example, when introducing the properties of PLCL, a comparison with other biological scaffold materials can be made

[Reply]: Dear reviewer, thank you for your comment. We have added the description as follows: “Current biodegradable materials for medical applications primarily encompass biodegradable metals and polymers. Among these, biodegradable polymers demonstrate superior suitability for cerebrovascular stents due to their enhanced flexibility and malleability, which facilitate better adaptation to the intricate anatomy of intracranial vasculature. As summarized in Table S2[63-67], poly-L-lactic acid (PLLA) represents the most widely utilized biodegradable polymer. Notably, the Igaki-Tamai stent, recognized as the first self-expanding PLLA-based resorbable stent implanted in humans, marked a significant milestone in this field [68]. However, PLLA exhibits certain limitations, including inherent brittleness and relatively slow degradation kinetics [68]. In contrast, poly(L-lactide-co- ϵ -caprolactone) (PLCL), a copolymer synthesized

from lactic acid (LA) and ϵ -caprolactone (CL), has emerged as an optimized alternative to PLLA. This advanced material offers several distinct advantages: (1) accelerated degradation rates[53], making it particularly suitable for applications requiring short-term support (typically 3-6 months); (2) enhanced flexibility and superior elongation at fracture; and (3) improved processing characteristics [69]. These properties not only address the limitations of PLLA but also render PLCL particularly advantageous for advanced manufacturing techniques, including 3D printing applications.” in the section of Intracranial stenting in the revised manuscript.

Table S2. Typical polymer based stents

Stent	Material	Animal model	Function	Functional Method	Reference
ABSORB BVS	PLLA	Human Coronary artery	Antiproliferative	PDLLA coating with Everolimus	2015[63]
DESolve NTx	PLA	Human Coronary artery	Anti-proliferative and anti-inflammatory	PLLA coating with Novolimus	2017[64]
MeRes100	PLLA	Human Coronary artery	Inhibiting neointima proliferation	PDLLA coating with Sirolimus	2011[65]
XINSORB	PLLA	Porcine coronary artery	Inhibiting neointima proliferation	PDLLA mixed with PLLA with Sirolimus	2012[66]
BRS	PLGA	Ovine femoral and profunda vessel	Self-expanding	Composite design: braided stent coated with an elastomer of PGCL and cross-linked	2018[67]
PLCL-MgSO ₄	PLCL	Rat internal carotid artery	Neuroprotection	Mix MgSO ₄ with PLCL	This study

PDLLA: Poly(D,L-lactic acid); PBAT: Poly(butyleneadipate-co-terephthalate); PDDA: Poly (dimethyl diallyl ammonium chloride); P(LA-TMC): poly(lactic acid-co-trimethylene carbonate); APTES: 3-amino-propyltrimethoxysilane (APTES); PLGA: poly(lactic-co-glycolic acid); PLLA: Poly-L-lactic Acid; PDLLA: Poly-D-L-Lactic acid; PGCL: poly(glycolide-co-caprolactone);

3.Lines 142 - 169: please clearly state or display the innovative characteristics in experimental design. For some complex results, such as the three stages of Mg²⁺ release, charts or diagrams can be used for more intuitive display, and the key characteristics of each stage can be emphasized in the text description.

[Reply]: Dear reviewer, thank you for your comment. We have added the state of the innovative characteristics in experimental design as follows: “To elucidate the Mg²⁺ release mechanism from PLCLxMS stents, we designed an experimental approach combining *in vitro* degradation studies with substrate degradation simulation. The Mg²⁺ release diffusion model, developed based on the *in vitro* immersion release profile, is presented in Fig. 3A. For verification purposes, we established a computational model that incorporates the proposed release mechanism and aligns with the diffusion model parameters” And “The key characteristics of each stage in the Mg²⁺ release profiles are summarized in Table S1.”in the section of In vitro degradation analysis in the revised manuscript.

Table S1. Key characteristics of Staged-Mg²⁺ release

Stage	Time	Rate of Mg ²⁺ release (mg·L ⁻¹ ·day ⁻¹)	Dissolved MgSO ₄	Mechanism
Burst	Hours	6.49-38.40	Surface	Dissolution
Fast	Weeks	0.15-4.08	Shallow layer	Diffusion
Stable	Months	0.02-0.30	Deep layer	Diffusion

4.Line 335: The rationality of the experimental sample size should be explained, including the basis for choosing the current sample size and whether the sample size is sufficient to support the research conclusions.

[Reply]: Dear reviewer, thank you for your comment. According to the ISO-10993 standard, five parallel samples from cellular experiments on biodegradable biomaterials are sufficient to support the study conclusions. And we have added the state: “Human umbilical vein endothelial cells (HUVECs), human artery smooth muscle cells (HASMCs), and HT-22 (a mouse central neuronal cell line) were selected to assess the

cytotoxicity of PLCLxMS composites according to the ISO-10993 standard[117].” in the section of *In vitro* cytotoxicity assays in the revised manuscript. And “To simulate ischemia/reperfusion injury in vitro, HT-22 cells underwent oxygen-glucose deprivation followed by reoxygenation (OGD/R) according to the ISO-10993 standard[118].” in the section of *In vitro* neuroprotection assessments in the revised manuscript.

5.Lines 648 - 655: please state the safety analysis of the materials, including the research on long - term safety and potential adverse reactions.

[Reply]: Dear reviewer, thank you for your comment. We have added the state of the safety analysis of the materials as follows: “The safety profile of PLCLxMS stents was evaluated through *in vitro* and *in vivo* biocompatibility assessments in this study. Initial *in vitro* investigations demonstrated well hemocompatibility and cytocompatibility. Subsequent one-month implantation studies in both rat and rabbit models revealed complete endothelialization and appropriate inflammatory responses of the stent. However, long-term safety considerations and potential side effects related to stent degradation warrant further investigation. The degradation mechanism of PLCLxMS stents involves two concurrent processes: magnesium sulfate dissolution and PLCL hydrolysis, both of which contribute to the maintenance of long-term safety.” in the section of Comparison with previous studies in the revised manuscript.

6.Line 858: the indicate removal after 2 hours, but there is no basis or timeline for fiber removal, and the corresponding clinical window's pathological mechanism is not provided.

[Reply]: Dear reviewer, thank you for your comment. It suggests that 2 hours can cause stable damage to both cortex and striatum relative to 1 hour, and therefore the 2 h injury model is more stable and is now widely used. And the clinical window's pathological mechanism corresponds to the beginning of the Hyperacute Stage referred to in Table S1. And we have modified the description as follows: “The filament was withdrawn after 2 hours to allow reperfusion of the ischemic hemisphere and get stable injury

model [120].” in the section of MCAO Model and Material Implantation in the revised manuscript.

7.Experimental scope: Although the study shows that the scaffold can effectively release magnesium ions at different stages, the long - term effects and the applicability to different types of stroke have not been sufficiently explored. In the future, the experimental scope can be expanded to further verify the long - term efficacy and safety of the scaffold. Further research on the degradation products of the scaffold after long - term implantation in the body and their impact on surrounding tissues, including possible inflammatory reactions and tissue repair processes, is needed.

[Reply]: Dear reviewer, thank you for your comment. Thank you very much for your suggestions and pointing out our shortcomings, we have added discussions to illustrate the long-term safety in question 5 and also added the future experimental directions of our research: “Given the clinical heterogeneity of ischemic stroke subtypes, future investigations should validate the neuroprotective efficacy of PLCLxMS stents across various stroke models, including large vessel occlusion, cardioembolic stroke, and small vessel disease [114]. Although our study has demonstrated the stent's safety and efficacy over a one-month period, several critical aspects require further investigation: (1) the temporal changes in mechanical integrity, (2) long-term in vivo degradation kinetics, and (3) the physiological impact of degradation products. These parameters remain to be fully elucidated and warrant comprehensive evaluation in future studies.” in the section of Comparison with previous studies in the revised manuscript.

8.Scaffold degradation characteristics: Although the degradation performance of the scaffold was mentioned, the long - term impact of the degradation rate on the compatibility with brain tissue was not discussed in depth. Especially whether the release mechanism of magnesium ions and the degradation process of the scaffold will have a negative impact on brain tissue is worth further

research.

[Reply]: Dear reviewer, thank you for your comment. Thank you very much for your suggestions and pointing out our shortcomings, we have added discussions the long - term impact of the degradation rate on the compatibility with brain tissue: “The degradation mechanism of PLCLxMS stents involves two concurrent processes: magnesium sulfate dissolution and PLCL hydrolysis, both of which contribute to the maintenance of long-term safety [102]. PLCL degradation yields lactic acid and caprolactone, which undergo metabolic conversion to water and carbon dioxide, subsequently eliminated through respiration and urinary excretion, demonstrating excellent long-term biocompatibility [103, 104]. Supporting this, Yuval Ramot et al. conducted comprehensive 52-week preclinical studies that validated the long-term biocompatibility and biodegradability of 70:30 PLCL [104]. While the degradation process may potentially induce mild inflammatory responses, foreign body reactions, or localized fibrosis due to lactic acid and caprolactone accumulation [105], the blood-brain barrier's functional integrity effectively regulates lactate and caprolactone diffusion [106]. This regulatory mechanism suggests that PLCL maintains favorable biocompatibility with cerebral tissue throughout its degradation period.” in the section of Comparison with previous studies in the revised manuscript.

9.Production cost and application feasibility of the scaffold: The cost issue of 3D - printed scaffold technology and the feasibility of its actual clinical application, especially the promotion for large - scale clinical application, were not discussed in detail in the article. As a new type of therapeutic method, how to reduce the cost and ensure its wide clinical application is an important issue that needs to be considered.

[Reply]: Dear reviewer, thank you for your comment. Thank you very much for your suggestions and pointing out our shortcomings, we have added discussions the 3D - printed scaffold technology and the feasibility of its actual clinical application: “The 3D printing methodology employed in this study demonstrates both cost-effectiveness

and manufacturing flexibility [69]. This technology has gained increasing recognition as a promising approach for fabricating personalized biodegradable polymer stents [70], particularly for small-batch production and highly customized applications. The PLCLxMS stents were fabricated using fused deposition modeling (FDM), a process wherein a heated nozzle precisely extrudes molten thermoplastic material along predetermined paths, with subsequent solidification forming the final product. Compared to conventional laser cutting techniques, FDM offers several advantages: (1) reduced material consumption, (2) lower manufacturing costs, and (3) enhanced capability for producing complex, patient-specific stent geometries, thereby ensuring the practical feasibility of PLCLxMS stent production. While 3D printing of pure PLCL stents has been established as a low-cost process [69], the fabrication of PLCLxMS stents utilizes identical equipment and processing parameters, with the modification of incorporating a PLCL-magnesium sulfate composite. Consequently, although the production cost of PLCLxMS stents is marginally higher than that of pure PLCL stents, it remains within a cost-effective range for medical device manufacturing.” in the section of Intracranial stenting in the revised manuscript.